# Control of pili synthesis and putrescine homeostasis in *Escherichia coli*

Iti Mehta[1†], Jacob B Hogins[1†], Sydney R Hall[1], Gabrielle Vragel[1], Sankalya Ambagaspitiye[1‡], Philippe E Zimmern[2], Larry Reitzer[1*]

[1]Department of Biological Sciences, The University of Texas at Dallas, Richardson, United States; [2]Department of Urology, University of Texas Southwestern Medical Center, Dallas, United States

## eLife Assessment

This **valuable** study presents an interesting analysis of the role of the polyamine precursor putrescine in the pili-dependent surface motility of a laboratory strain of *Escherichia coli*. The overall data **convincingly** demonstrate a role in this case. This study presents interesting findings for those studying uropathogenic bacteria, and those studying bacterial polyamine function.

**\*For correspondence:**
reitzer@utdallas.edu

[†]These authors contributed equally to this work

**Present address:** [‡]Department of Biosystems Technology, Faculty of Technology, University of Sri Jayewardenepura, Homagama, Sri Lanka

**Competing interest:** The authors declare that no competing interests exist.

**Abstract** Polyamines are biologically ubiquitous cations that bind to nucleic acids, ribosomes, and phospholipids and, thereby, modulate numerous processes, including surface motility in *Escherichia coli*. We characterized the metabolic pathways that contribute to polyamine-dependent control of surface motility in the commonly used strain W3110 and the transcriptome of a mutant lacking a putrescine synthetic pathway that was required for surface motility. Genetic analysis showed that surface motility required type 1 pili, the simultaneous presence of two independent putrescine anabolic pathways, and modulation by putrescine transport and catabolism. An immunological assay for FimA—the major pili subunit, reverse transcription quantitative PCR of *fimA*, and transmission electron microscopy confirmed that pili synthesis required putrescine. Comparative RNAseq analysis of a wild type and Δ*speB* mutant which exhibits impaired pili synthesis showed that the latter had fewer transcripts for pili structural genes and for *fimB* which codes for the phase variation recombinase that orients the *fim* operon promoter in the ON phase, although loss of *speB* did not affect the promoter orientation. Results from the RNAseq analysis also suggested (a) changes in transcripts for several transcription factor genes that affect *fim* operon expression, (b) compensatory mechanisms for low putrescine which implies a putrescine homeostatic network, and (c) decreased transcripts of genes for oxidative energy metabolism and iron transport which a previous genetic analysis suggests may be sufficient to account for the pili defect in putrescine synthesis mutants. We conclude that pili synthesis requires putrescine and putrescine concentration is controlled by a complex homeostatic network that includes the genes of oxidative energy metabolism.

## Introduction

Polyamines are flexible aliphatic cations that are found in virtually all organisms. *E. coli* contains putrescine (1,4-diaminobutane) and lesser amounts of spermidine (N-(3-aminopropyl)–1,4-diaminobutane), and cadaverine (1,5-diaminopentane) (*Tabor and Tabor, 1984*). Both the pathways and enzymes of polyamine synthesis are redundant (*Figure 1*). Strains that lack eight of these genes do not contain detectable polyamines, grow slowly aerobically, and do not grow anaerobically (*Chattopadhyay et al., 2009*). Polyamines interact with nucleic acids and phospholipids, and can affect chromosome and ribosome structure, ribosome-mRNA interactions, and protein and nucleic acid elongation rates

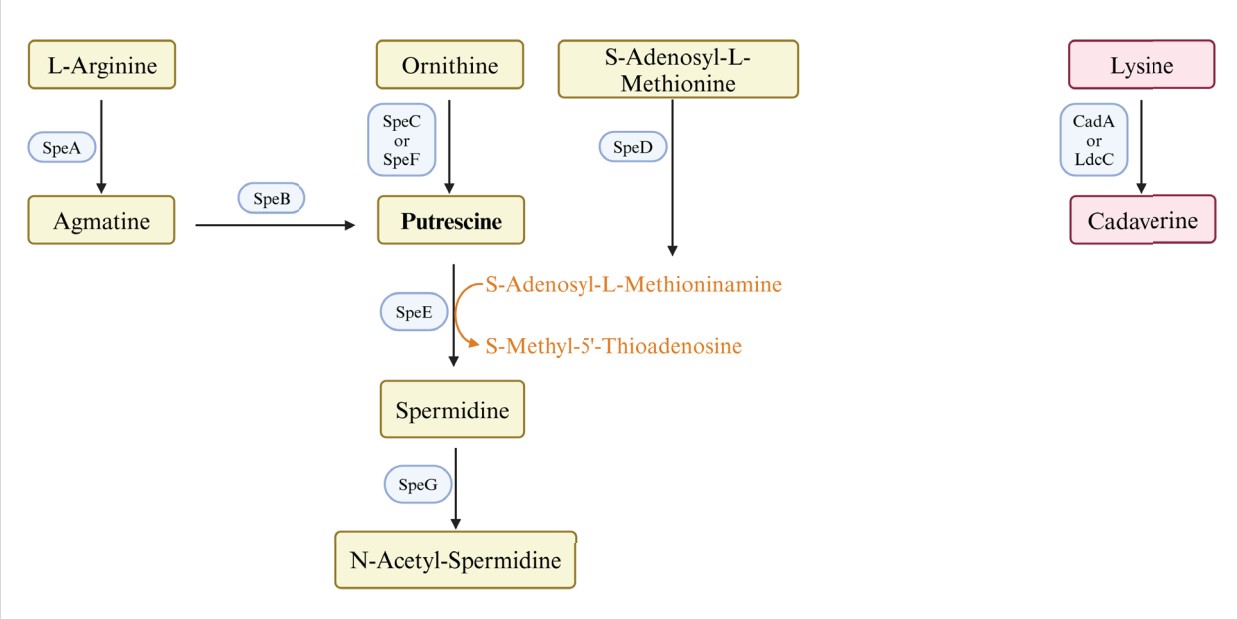

**Figure 1.** Pathways and enzymes of polyamine synthesis.

(*Igarashi and Kashiwagi, 2000*). Polyamines modulate, but are not required for, protein synthesis (*Igarashi and Kashiwagi, 2015*) and expression of hundreds of genes (*Yoshida et al., 2004*). A major mechanism of polyamine-dependent regulation is translational control of genes for 18 major transcription regulators: four of the seven σ factors – $\sigma^{18}$ (FecI), $\sigma^{24}$ (RpoE), $\sigma^{38}$ (RpoS), and $\sigma^{54}$ (RpoN); three histone-like DNA-binding proteins (Fis, H-NS, and StpA); and CpxR, Cya, Cra, and UvrY (*Igarashi and Kashiwagi, 2015*; *Yoshida et al., 2004*).

Polyamines control flagella-dependent surface motility of *Proteus mirabilis* and pili-dependent surface motility (PDSM) of *E. coli*: the latter has been reported to require the polyamine spermidine (*Armbruster et al., 2013*; *Kurihara et al., 2009*). While reconstructing putrescine-deficient strains to characterize the basis for the anaerobic growth defect, we noticed that a mutant unable to synthesize spermidine moved normally, while mutants lacking enzymes of putrescine synthesis moved poorly. We characterized the role of polyamines on *E. coli* surface motility and pili synthesis by genetic analysis, ELISA assays, RT-qPCR, electron microscopy, and RNAseq. We showed that PDSM and pili synthesis required two independent pathways of putrescine synthesis, but did not require spermidine, and was modulated by putrescine transport and catabolism. Results from an RNAseq analysis are consistent with multiple mechanisms of putrescine-dependent control, and, unexpectedly, suggest a putrescine homeostatic network that rewires metabolism to maintain intracellular putrescine.

## Results

### W3110 surface motility requires pili

An assessment of surface motility for *E. coli* K-12 and eight derivatives showed five strains covered a plate in 12–18 hr, while four did not traverse a plate in 36 hr (*Ambagaspitiye et al., 2019*). The slow-moving strains generated genetically stable fast-moving variants which suggests that the former were ancestral. We chose our lab strain of W3110 for further study because (a) it exhibited the least variability in spreading diameter and generated fewer fast-moving variants, and (b) we had previously analyzed this strain for the energy requirements for surface motility (*Sudarshan et al., 2021*). We refer to our slow-moving lab strain hereafter as W3110 but not all W3110 strains move similarly: the W3110 from the genetic stock center is a fast-moving variant (*Ambagaspitiye et al., 2019*).

Several results suggested that W3110 requires pili for surface motility. First, the deletion of *fimA*, which codes for the major type 1 pili subunit, abolished the oscillatory pattern and reduced, but did not eliminate, surface motility, but the deletion of *fliC* which encodes the major flagellum subunit did

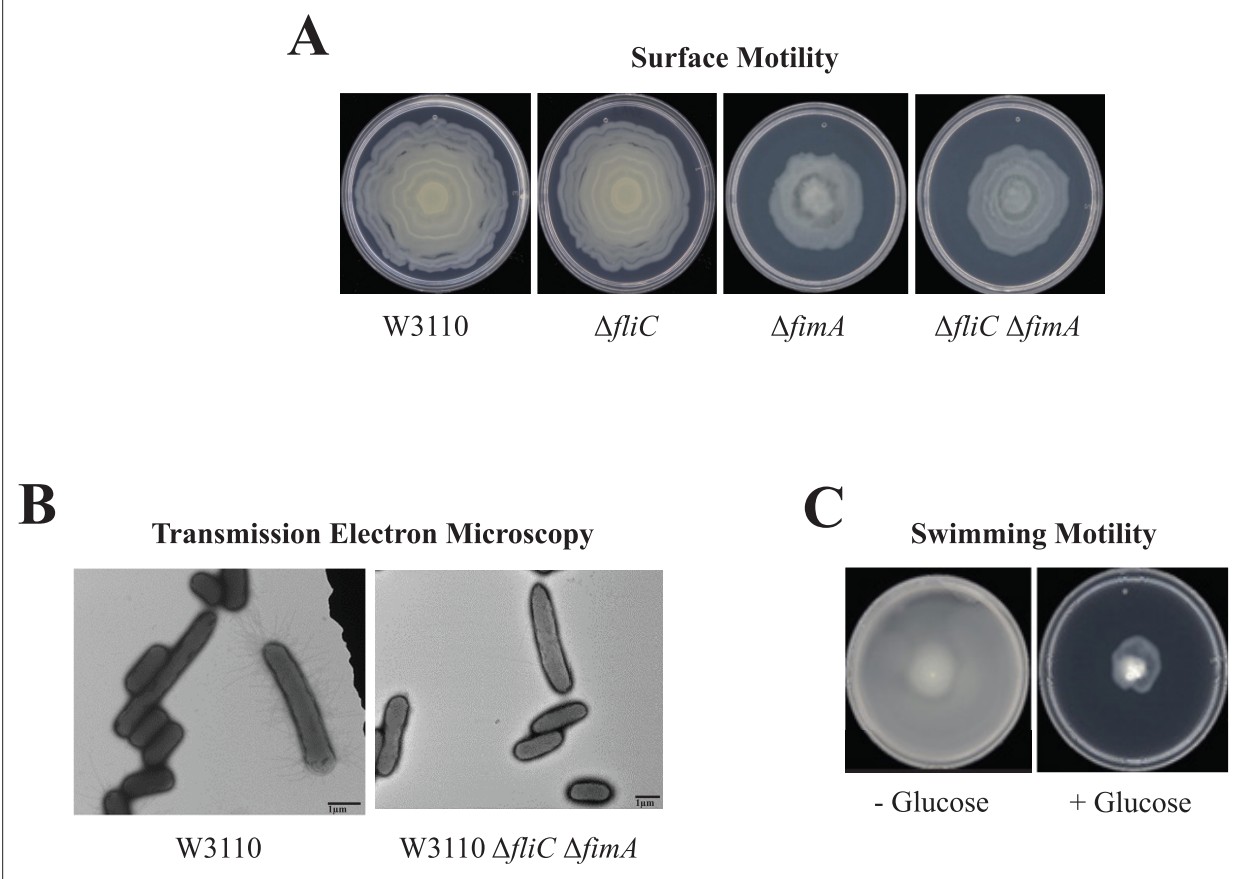

**Figure 2.** Surface motility of W3110. (**A**) Surface motility of parental and derivative strains lacking the major subunits of the flagella (FliC), pili (FimA), or both. (**B**) Transmission electron microscopy (TEM) images of cells of W3110 and W3110 ΔfliC ΔfimA were taken from the movement's edge directly from surface motility plates. (**C**) Swimming motility with and without 0.5% glucose.

not affect surface motility (*Figure 2A*). Second, electron microscopy confirmed that W3110 isolated from surface motility plates possessed pili (*Figure 2B*). None of >500 cells expressed flagella. A mixed population of elongated (3–4 µm) and non-elongated (<2 µm) cells were visible, and pili were mostly associated with elongated cells. Finally, W3110 surface motility requires glucose (*Sudarshan et al., 2021*) which should prevent flagella synthesis via inhibition of cyclic AMP synthesis (*Yokota and Gots, 1970*). We confirmed that glucose suppressed swimming motility which implies suppression of flagella synthesis in W3110 during surface motility assays (*Figure 2C*). A W3110 derivative with deletions of both *fliC* and *fimA* still exhibited outward movement on a surface motility plate (*Figure 2A*), but electron microscopy showed the absence of appendages (*Figure 2C*). In summary, genetics, electron microscopy, and the glucose requirement show that W3110 surface motility requires pili.

Results from W3110 sequencing found an *IS1* insertion in *fimE* which could explain its greater movement that is less variable and greater genetic stability. The FimB and FimE recombinases are the major components of phase variation that determine the orientation of the promoter that initiates transcription of the *fimAICDFGH* operon which codes for the pili structural proteins. The FimB recombinase favors the productive ON orientation, while the more active FimE recombinase favors the unproductive OFF orientation (*Conway et al., 2023*; *Kelly et al., 2006*; *O'Gara and Dorman, 2000*). The insertion in *fimE* is the likely basis for the relative stability of W3110.

## PDSM required two independent putrescine synthesis pathways

We examined PDSM requirements for polyamine anabolic pathways. A major putrescine biosynthetic pathway is ornithine decarboxylation either by the constitutive SpeC or the low-pH inducible SpeF. The ΔspeC and ΔspeF mutants moved as well as the parental strain, but a ΔspeC ΔspeF double mutant moved less well (*Figure 3*). SpeA (arginine decarboxylase) and SpeB (agmatinase) catalyze an

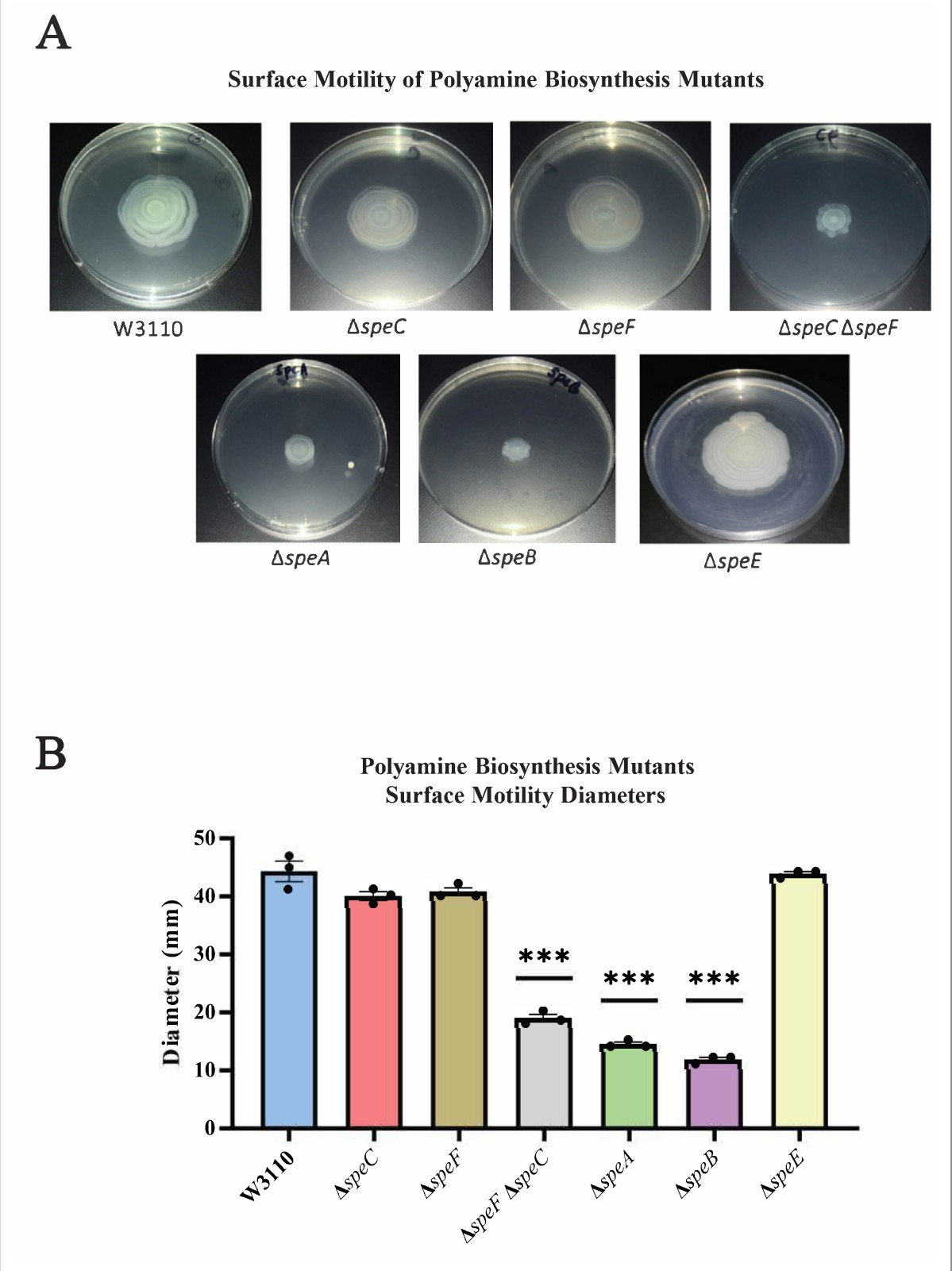

**Figure 3.** Genetics of surface motility. (**A**) Pili-dependent surface motility (PDSM) of mutants with defects in polyamine anabolic genes. All assays were performed in triplicate, and representative images are shown. (**B**) Diameter of surface movement of polyamine mutants after 36 hr. Error bars represent standard deviations for three independent replicates. Statistical analysis was performed using the Dunnett test of significance: *p<0.05; **p<0.01; ***p<0.001. In this figure the Δ*speB* mutant was IM26.

*Figure 3 continued on next page*

*Figure 3 continued*

The online version of this article includes the following figure supplement(s) for figure 3:

**Figure supplement 1.** Growth and swimming of polyamine anabolic mutants.

alternate two-step putrescine synthetic pathway (*Figure 1*; *Tabor and Tabor, 1985*): deletion of either gene reduced surface motility (*Figure 3*). The ∆*speA*, ∆*speB*, and ∆*speC* ∆*speF* strains grew as well as the parental strain and had normal swimming motility (*Figure 3—figure supplement 1*). A ∆*speE* mutant, which lacks the enzyme for spermidine synthesis, moved on a surface as well as the parental strain (*Figure 3*).

Supplemental putrescine at 1 mM restored the surface motility of a *speB* mutant (*Figure 4A*) and a *speA* mutant (not shown). Spermidine did not restore wild-type movement and, also, caused an unusual pattern of movement in the parental strain (*Figure 4A*). These exogenous concentrations of putrescine and spermidine had no effect on growth rates in a liquid motility medium (not shown). In summary, PDSM required the simultaneous presence of two independent pathways of putrescine synthesis but did not require spermidine.

## Loss of putrescine transport systems affected surface motility

*E. coli* has several polyamine transport systems and one has been reported to be required for surface motility (*Igarashi et al., 2001*; *Kurihara et al., 2011*). Mutants lacking the PlaP and PotF putrescine transporters had 50% and 40% reduced surface motility, respectively, and lost the concentric ring pattern (*Figure 5A*). Loss of the PotE and PuuP transporters affected neither the movement diameter (*Figure 5B*) nor concentric ring formation (*Figure 5A*). We note that *potF* and *plaP* are more highly expressed than *potE* and *puuP* (85, 64, 17, and 6 counts per million transcripts, respectively) for W3110 grown in a liquid motility medium (*Hogins et al., 2023a*). We conclude that extracellular putrescine and its transport contribute to PDSM.

## Putrescine catabolism affected surface motility

Because putrescine catabolism modulates intracellular putrescine concentrations (*Schneider et al., 2013*; *Schneider and Reitzer, 2012*), we examined PDSM in putrescine catabolic mutants. The major putrescine catabolic pathway is initiated with putrescine glutamylation (*Figure 6A*, right column), and a second pathway initiates with putrescine deamination (*Figure 6A*, left column) (*Schneider and Reitzer, 2012*). Deletion of either *puuA* or *patA* which codes for the first enzymes of their respective pathways did not affect surface motility (*Figure 6B*). A double mutant moved less well which was unexpected because the proposed higher intracellular putrescine should have stimulated PDSM. Consistent with the stimulatory effect of putrescine, loss of *puuR* which codes for the repressor of PuuA-initiated pathway genes (*Schneider and Reitzer, 2012*), impaired PDSM (*Figure 6B*). One possible explanation to reconcile these seemingly contradictory results is that an optimal polyamine concentration stimulates PDSM, and a high concentration is inhibitory. The next section describes experiments to test this possibility.

## High putrescine reduced the expression of pili genes

High putrescine inhibits the translation of some mRNAs, and high spermidine, a product of putrescine metabolism, inhibits growth (*Fukuchi et al., 1995*; *Sakamoto et al., 2020*). If the phenotype of the ∆*patA* ∆*puuA* double mutant results from spermidine toxicity, then loss of spermidine synthase (SpeE) should reverse the phenotype. This prediction was not met, which argues against spermidine toxicity (*Figure 6C*). For the *speB* mutant, 1 mM putrescine stimulated PDSM, and 4 mM putrescine was slightly inhibitory (*Figure 7A and B*). The concentric ring pattern was observed with 1 mM, but not 4 mM putrescine. RT-qPCR to test transcriptional regulation of pili expression by putrescine showed optimal *fimA* transcription at 1 mM putrescine in the *speB* mutant (*Figure 7C*). Indirect enzyme-linked immunosorbent assays (ELISAs) against FimA suggested that the level of FimA in the *speB* mutant at 1 mM putrescine was higher than with 4 mM putrescine, but the difference was not statistically significant (*Figure 7D*). Variation of supplemental putrescine did not affect FimA in the parental strain (*Figure 7C and D*). Transmission electron microscopy (TEM) showed that pili expression was highest for the *speB* mutant at 1 mM putrescine (*Figure 8*), moderate at 0.1- and

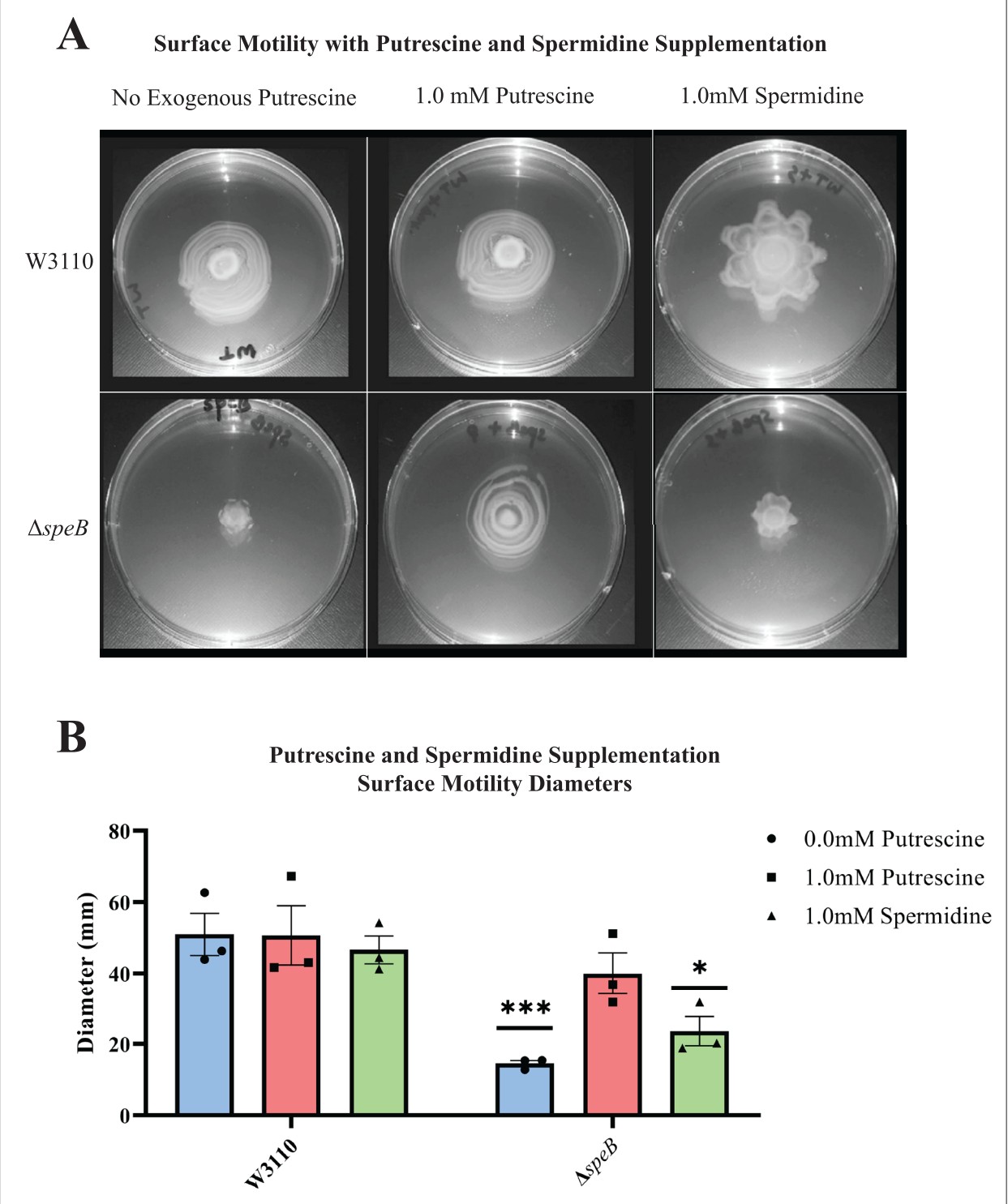

**Figure 4.** Nutritional supplementation of Δ*speB* mutant. (**A**) Putrescine and spermidine supplementation of wild-type and Δ*speB* strains. (**B**) Diameter of surface movement of polyamine mutants after 36 hr. Error bars represent standard deviations for three independent replicates. Statistical analysis was performed using the Sadik test of significance: *p<0.05; **p<0.01; ***p<0.001. All assays were performed in triplicates and representative images are shown. In this figure the Δ*speB* mutant was IM26.

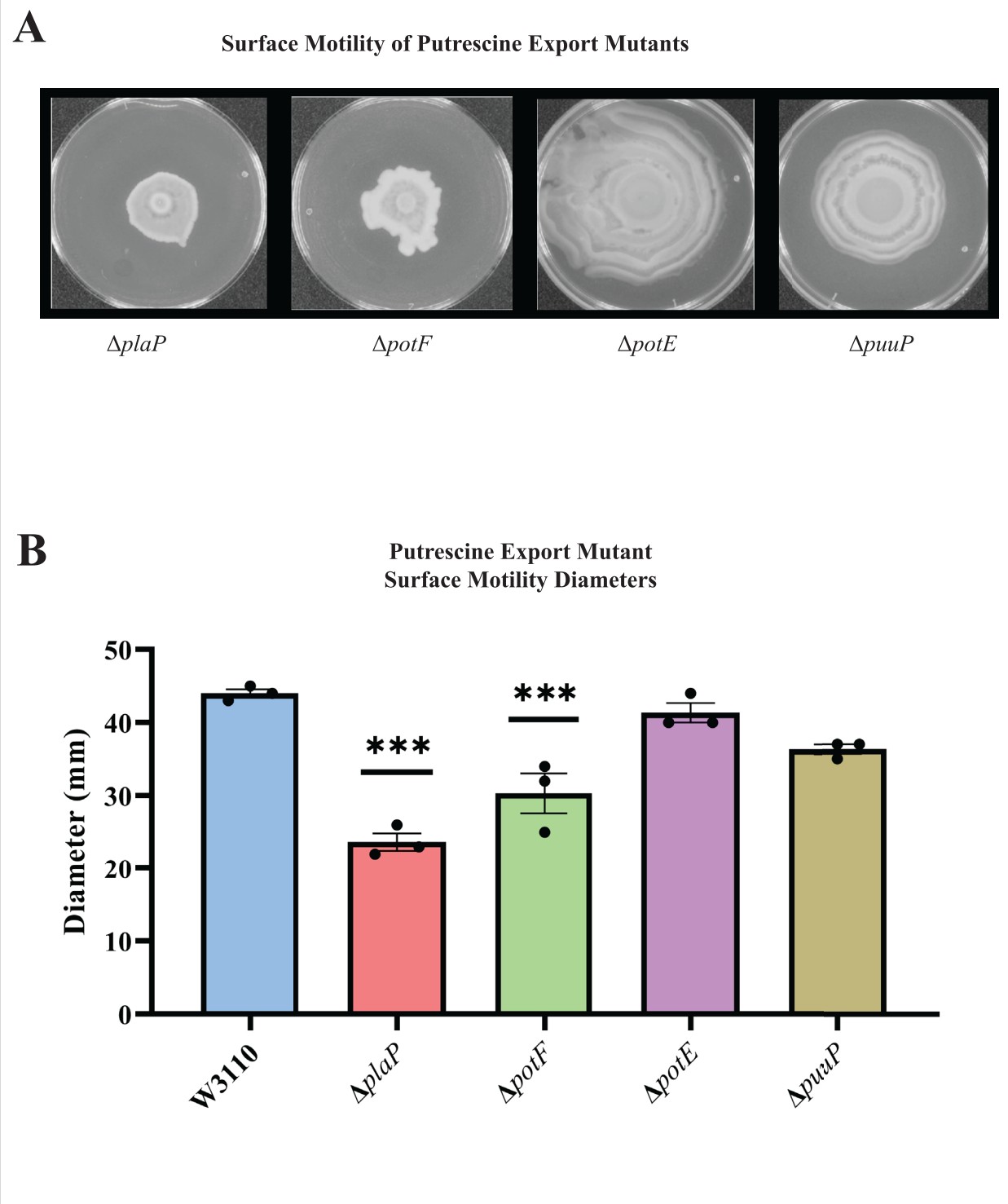

**Figure 5.** Putrescine transport and surface motility. (**A**) Representative images of surface motility for strains defective in putrescine transport genes. (**B**) Diameter of surface movement of polyamine mutants after 36 hr. Error bars represent standard deviations for three independent replicates. Statistical analysis was performed using the Dunnett test of significance: *p<0.05; **p<0.01; ***p<0.001. All assays were performed in triplicate and representative images are shown.

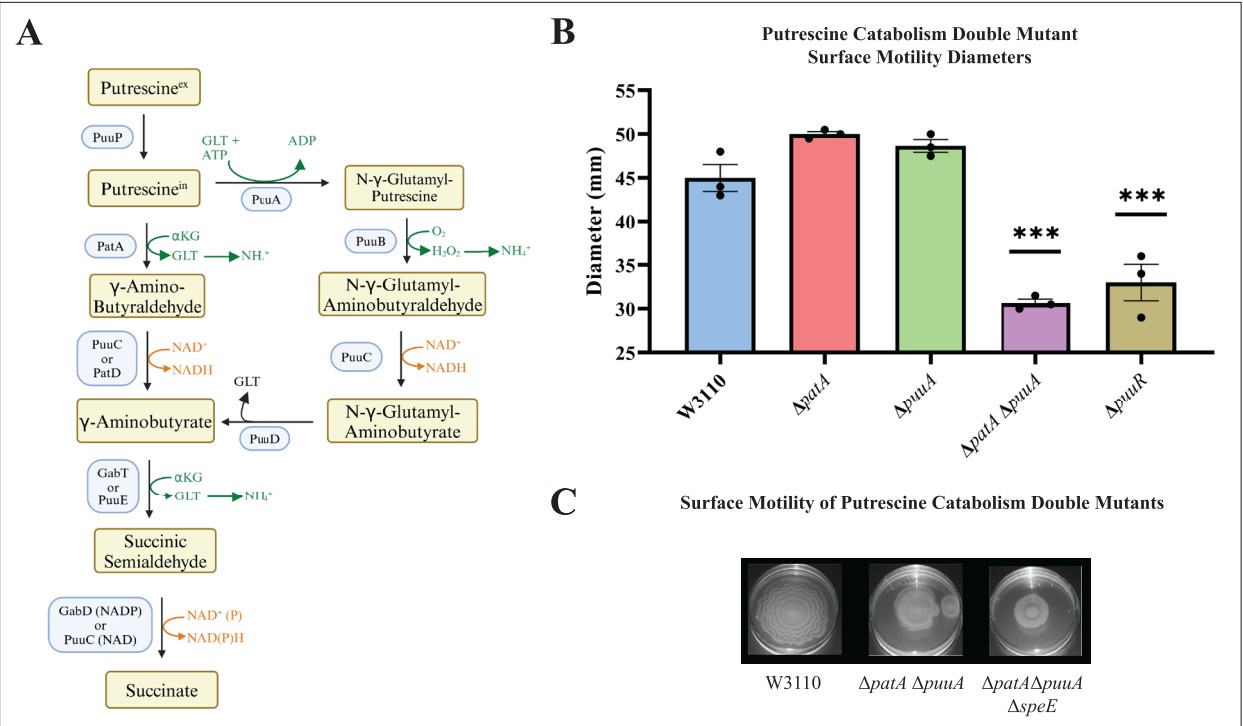

**Figure 6.** Putrescine catabolism and surface motility. (**A**) Pathways and enzymes of putrescine catabolism. (**B**) Motility diameter of putrescine catabolic mutants after 36 hr. Error bars represent standard deviations for three independent replicates. Statistical analysis was performed using the Dunnett test of significance: *$p<0.05$; **$p<0.01$; ***$p<0.001$. All assays were performed in triplicates and representative images are shown. (**C**) Effect of loss of *speE* on the *patA puuA* catabolic double mutant.

4 mM putrescine, and undetectable without putrescine. The cells grown with 4 mM putrescine were shorter and thinner, which suggests stress, possibly mimicking osmotic stress. From these results and those in the previous section, we conclude that pili expression requires an optimal putrescine concentration.

## RNAseq analysis identifies *fim* operon expression and energy metabolism as targets of putrescine control

RNAseq analysis was used to further confirm or determine whether putrescine affected (a) transcription of the *fim* operon that codes for the pili structural genes, (b) *fim* operon promoter orientation, i.e., phase variation, or (c) energy metabolism: surface motility of W3110 has been shown to require glucose metabolism and oxidative phosphorylation (*Sudarshan et al., 2021*).

### RNAseq analysis

We grew parental W3110 and its Δ*speB* derivative with and without 1 mM putrescine. $R^2$ values of pairwise comparisons showed that the transcriptome of the *speB* mutant grown without putrescine differed from the other transcriptomes which were similar (summarized in *Table 1* and shown graphically in *Supplementary file 1*). A multidimensional scaling plot and a plot of the 100 most variable genes visualize these comparisons (*Figure 9A and B*). When compared to the parental strain, the *speB* mutant grown without putrescine had 310 downregulated genes and 159 upregulated genes with at least fourfold differential expression and FDR <0.05. Transcripts for the putrescine-induced *puuAP* and *puuDRCBE* operons, which specify genes of the major putrescine catabolic pathway, were reduced from 1.6- to 14-fold (FDR ≤0.02) in the *speB* mutant (*Supplementary file 2*). Because higher intracellular putrescine results in higher catabolic gene expression, these results imply lower intracellular putrescine in the Δ*speB* mutant.

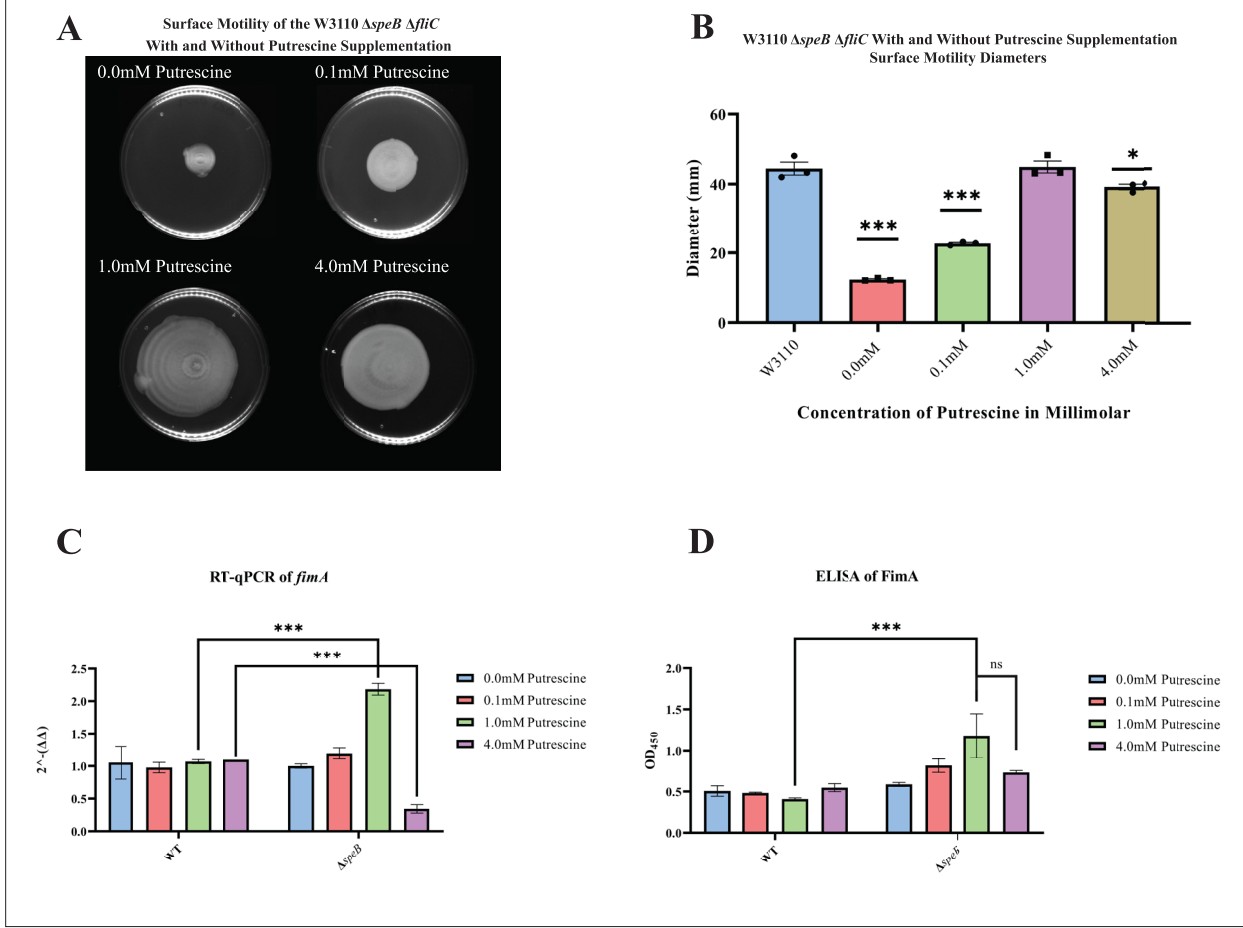

**Figure 7.** Surface motility and pili expression in W3110 and Δ*speB*. (**A**) W3110 Δ*speB* surface motility with 0.0, 0.1, 1.0, and 4.0 mM exogenous putrescine. All assays were performed in triplicate. Representative images are shown. (**B**) Average diameter of three separate surface motility plates. The parental strain without putrescine is shown for reference. Significance was determined by comparing the diameters of the Δ*speB* mutants in the different concentrations of putrescine compared to the parental W3110. One-way ANOVA was used with Dunnett hypothesis testing to determine p values, ***p<*0.001*. (**C**) Reverse transcriptase-quantitative PCR using primers targeting the *fimA* gene from cells grown in 0.0, 0.1, 1.0, and 4.0 mM of supplemented putrescine. Double deltas were generated by normalizing the parental and the Δ*speB* mutant RNA libraries to *rpoD* t hen comparing *fimA* expression. Two-way ANOVA was used to determine significance followed by Sadik hypothesis testing. ***p<0.001. (**D**) Enzyme linked immunosorbent assays using antibodies targeting pili (FimA) from cells grown with 0.0, 0.1, 1.0, and 4.0 mM of supplemented putrescine. Two-way ANOVA was used to determine significance followed by FDR adjusting. ***p<0.001. In this figure the Δ*speB* mutant was J15.

### Transcription of the *fim* operon is reduced in the Δ*speB* mutant

Deletion of *speB* reduced transcripts for genes of the *fimA* operon (**Figure 10A**). Numerous transcription factors activate the *fim* operon (**Schwan, 2011**; **Karp et al., 2018**), and, of these, loss of *speB* reduced *hns* transcripts, and increased *fis*, *lrp*, and *qseB* transcripts, but had no effect on *ihfA* and *ihfB* (**Figure 10B**). Loss of *hns*, *ihfA*, and *lrp* impaired PDSM (**Figure 10C and D**), which confirms the requirement for their products for pili synthesis in our strain background.

### Putrescine does not control *fim* operon phase variation in W3110

The phase ON-favoring FimB and phase OFF-favoring FimE recombinases control the orientation of the *fim* operon promoter. Our version of W3110 has an insertion in *fimE* which means that FimB is the major factor that controls phase variation. Loss of *speB* reduced transcripts from *fimB* which could account for loss of motility. PCR analysis with specific primers can determine the *fim* operon promoter orientation (**Figure 11A**) and showed that loss of *speB*, loss of *hns*, or the presence of putrescine did not alter the ratio of ON to OFF (**Figure 11B**). This experiment determines the steady state phase ON to OFF ratio, which less FimB might not affect if FimB is in excess. In strains with FimE, the rate of

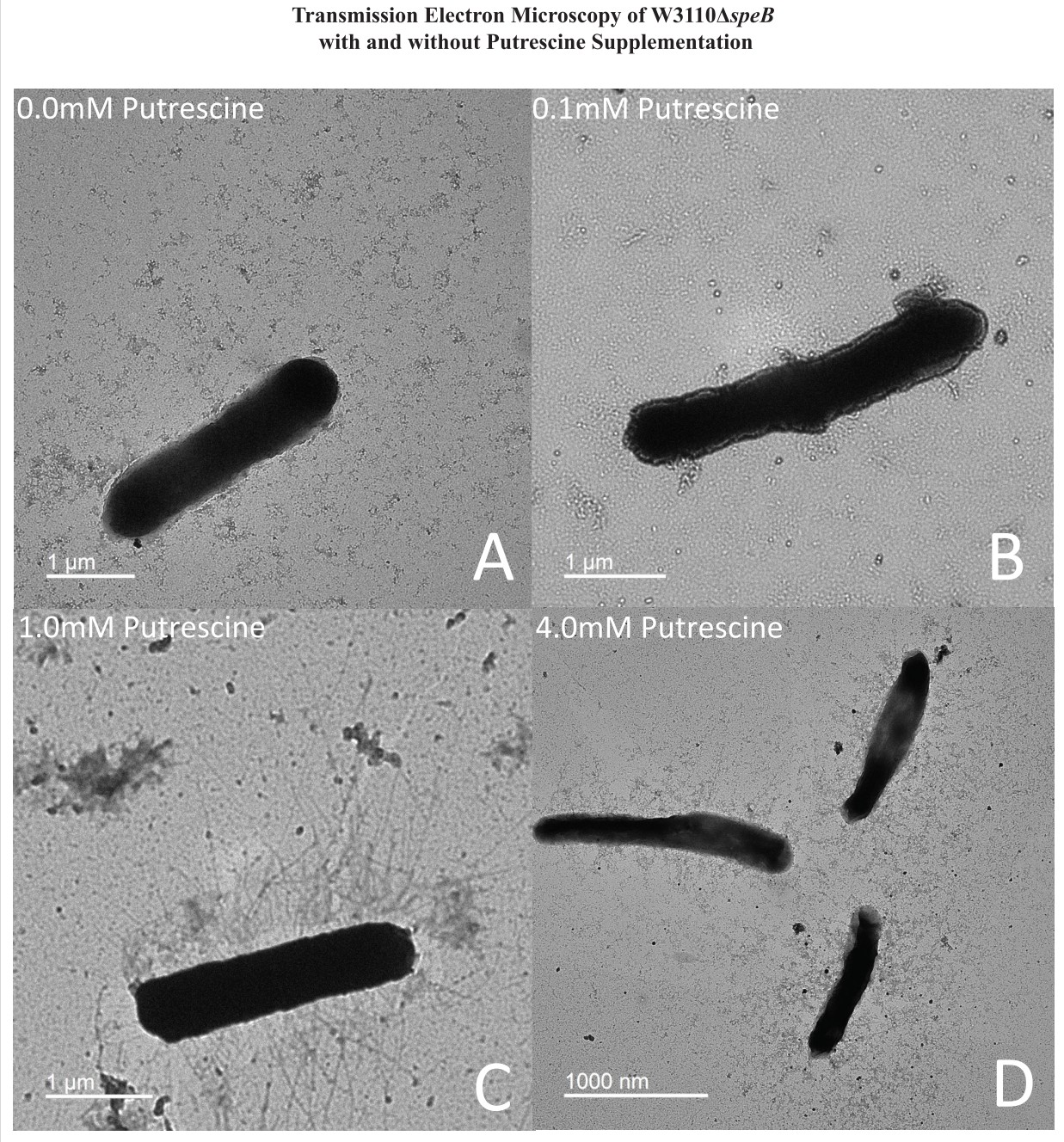

**Figure 8.** Transmission electron micrographs of Δ*speB* mutant cells after surface motility. Representative images are shown and cells were removed from motility plates with the following exogenous putrescine concentrations: (**A**) none, (**B**) 0.1 mM, (**C**) 1.0 mM, and (**D**) 4.0 mM. No pili were observed with 0 and 0.1 mM putrescine. Optimal pili production was observed with 1.0 mM putrescine. The bar represents one micron. In this figure the Δ*speB* mutant was J15.

phase ON orientation formation will be much slower, and even small changes in FimB activity could conceivably affect the switch orientation. We conclude that putrescine does not control phase variation, at least in our strain of W3110.

## Evidence for a putrescine homeostatic network

Altered transcript levels in the *speB* mutant suggest compensatory mechanisms for low putrescine (*Supplementary file 2* for transcript differences for all genes, *Table 2* for selected genes, and

**Table 1.** Pairwise statistical comparisons of transcriptomes.
Abbreviation: putr is putrescine.

| Strain 1 | Strain 2 | $R^2$ |
| --- | --- | --- |
| Wild-type (1 mM putr) | wild-type (0 mM putr) | 0.950 |
| Wild-type (1 mM putr) | Δ*speB* (1 mM putr) | 0.954 |
| Δ*speB* (0 mM putr) | Δ*speB* (1 mM putr) | 0.820 |
| Δ*speB* (0 mM putr) | wild-type (0 mM putr) | 0.796 |
| Δ*speB* (0 mM putr) | wild-type (1 mM putr) | 0.786 |

*Figure 12* for a diagrammatic representation of transcript differences). The *speB* mutant had more transcripts from (a) genes of arginine and ornithine carboxylase, (b) all genes of ornithine and arginine synthesis from glutamate, and (c) genes for three separate putrescine transport systems; and fewer transcripts from the *sap* operon which codes for a putrescine exporter (*Sugiyama et al., 2016*) and *rpmH* and *rpsT*, whose products inhibit ornithine and arginine decarboxylase (*Keseler et al., 2017*). The net effect of these changes should be to increase intracellular putrescine (solid arrows in *Figure 12*). In other words, these changes are compensatory for low putrescine. In addition to these changes, the *speB* mutant had 60-fold and sevenfold more transcripts for *mgtA* and *phoBR* which code for the major magnesium transporter and the regulators of phosphate assimilation, respectively. These changes may also be compensatory and are discussed below.

## Putrescine affects energy metabolism which could account for loss of motility

Loss of *speB* affected transcripts for genes of the major energy metabolism pathways (*Table 2*): fewer transcripts for genes coding for all citric acid cycle enzymes, NADH dehydrogenase I (the entire *nuo* operon), cytochrome oxidase (the *cyo* operon), and several genes of menaquinone (the *menFDHBCE* operon), ubiquinone, and heme synthesis. The *speB* mutant also had fewer transcripts for all genes of iron acquisition, which is consistent with diminished oxidative energy generation. A *speB* mutant had fewer transcripts for *ptsH* (Hpr of the phosphotransferase system of carbohydrate transport), *ptsG* (enzyme IIBC component of glucose transport), *gltA* (citrate synthase), *sdhA* (a succinate dehydrogenase subunit), and *sucC* (a succinyl-CoA synthetase subunit), and their loss has been shown to either impair or eliminate W3110 surface motility (*Sudarshan et al., 2021*). The cumulative effect of these differences could account for the motility defect in the *speB* mutant.

## Discussion

Several aspects of polyamine metabolism affect PDSM in *E. coli*. Strains with loss of one or two polyamine synthesis genes — *speA* and *speB* single mutants and a *speC speF* double mutant — resulted in PDSM defects. This observation is remarkable because nine enzymes and several pathways are involved in polyamine synthesis. Our putrescine synthesis mutants had no growth defect which is not unexpected since deletions of eight biosynthetic genes were required to show a strong phenotype: impaired aerobic growth and elimination of anaerobic growth (*Chattopadhyay et al., 2009*). Strains with defects in the two most active putrescine transport systems had altered PDSM. The specific defects suggest the involvement of extracellular putrescine and arginine in intracellular putrescine homeostasis. The PDSM defect of putrescine transport mutants not only implicates extracellular putrescine but also suggests secretion of putrescine which is not added to the medium. The PDSM defect of *speA* and *speB* mutants implicates extracellular arginine because SpeA is mostly periplasmic (*Tabor and Tabor, 1969*). Defective putrescine catabolism also affected PDSM. The role of both putrescine synthesis and degradation suggests rapid adjustments to fluctuating putrescine concentrations and environmental conditions: putrescine catabolism is a core response to several stresses (*Schneider et al., 2013*).

Results from an RNAseq analysis showed that transcription of the *fim* operon was reduced in the *speB* mutant. Of the known transcription factors that control the *fim* operon, only the gene for

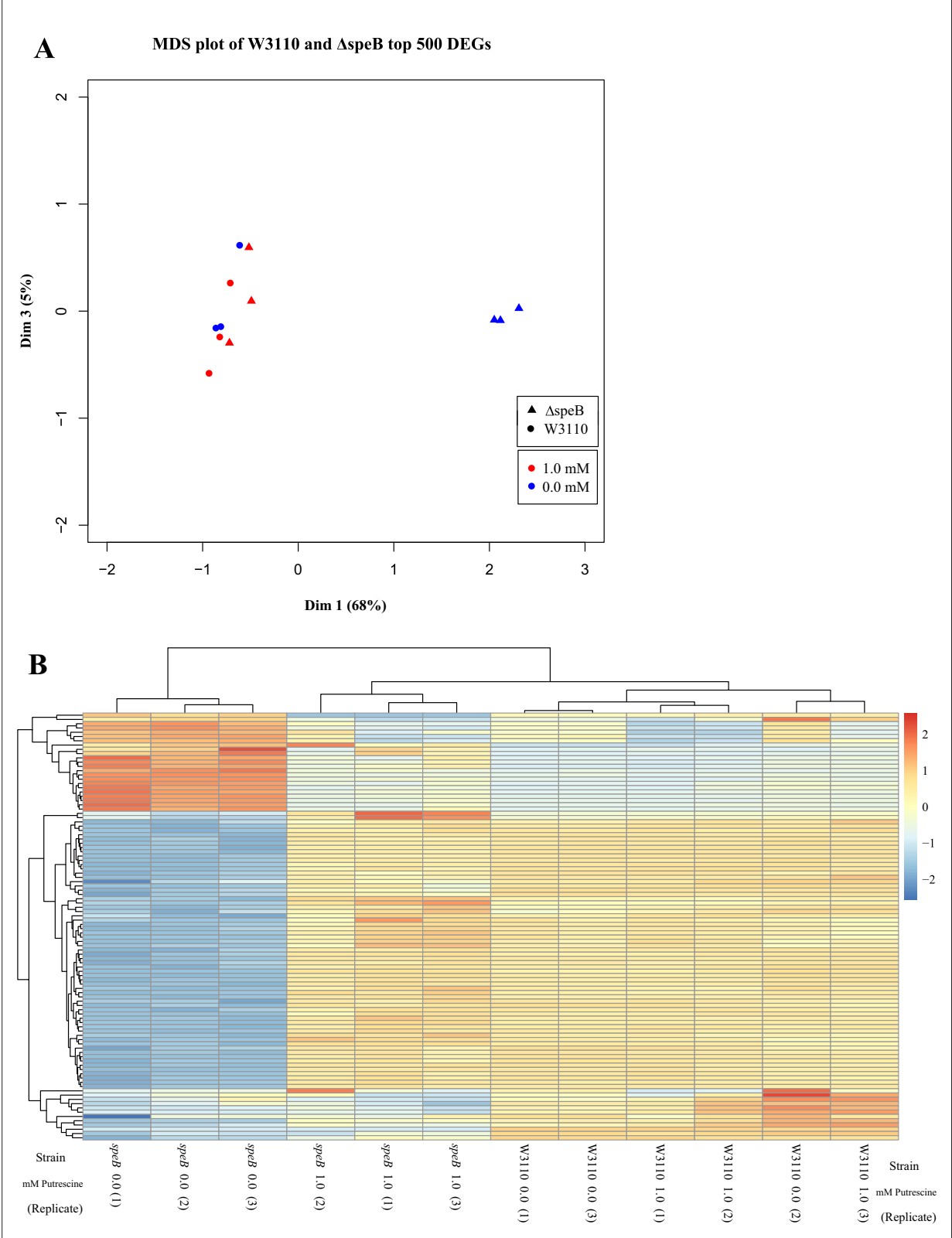

**Figure 9.** Visual representation of the results from transcriptomic sequencing of W3110 and the Δ*speB* mutant's gene expression in media with and without 1.0 mM putrescine. (**A**) Multidimensional scaling plot of the parental and Δ*speB* mutant transcriptomes. When grown with 1.0 mM putrescine (red), the Δ*speB* mutant transcriptome (triangles) are nearly identical to the parental transcriptomes when grown without and with putrescine (blue and red circles, respectively). When grown without putrescine, the Δ*speB* mutant transcriptome (blue triangles) is greatly skewed from transcriptomes of the

*Figure 9 continued on next page*

*Figure 9 continued*

parental strain and the Δ*speB* mutant grown with putrescine. (**B**) Heatmap of the top 100 most variable genes further demonstrates the distinctiveness of the Δ*speB* mutant grown without putrescine and the similarities of the Δ*speB* mutant transcriptome when grown with 1.0 mM putrescine supplementation and the parental grown with or without putrescine. In this figure the Δ*speB* mutant was J15.

H-NS had fewer transcripts. In contrast, transcripts for other relevant transcription factor genes were elevated in the *speB* mutant. Our results are consistent with H-NS mediating putrescine-dependent control, but we cannot exclude a general regulatory dysfunction mediated by altered nucleoprotein complexes at the *fim* operon promoter. The elevated transcripts for some transcription factors may be inhibitory.

The transcriptomic results showed that the *speB* mutant had more transcripts for genes whose products will maintain intracellular putrescine, which suggests compensatory mechanisms for low

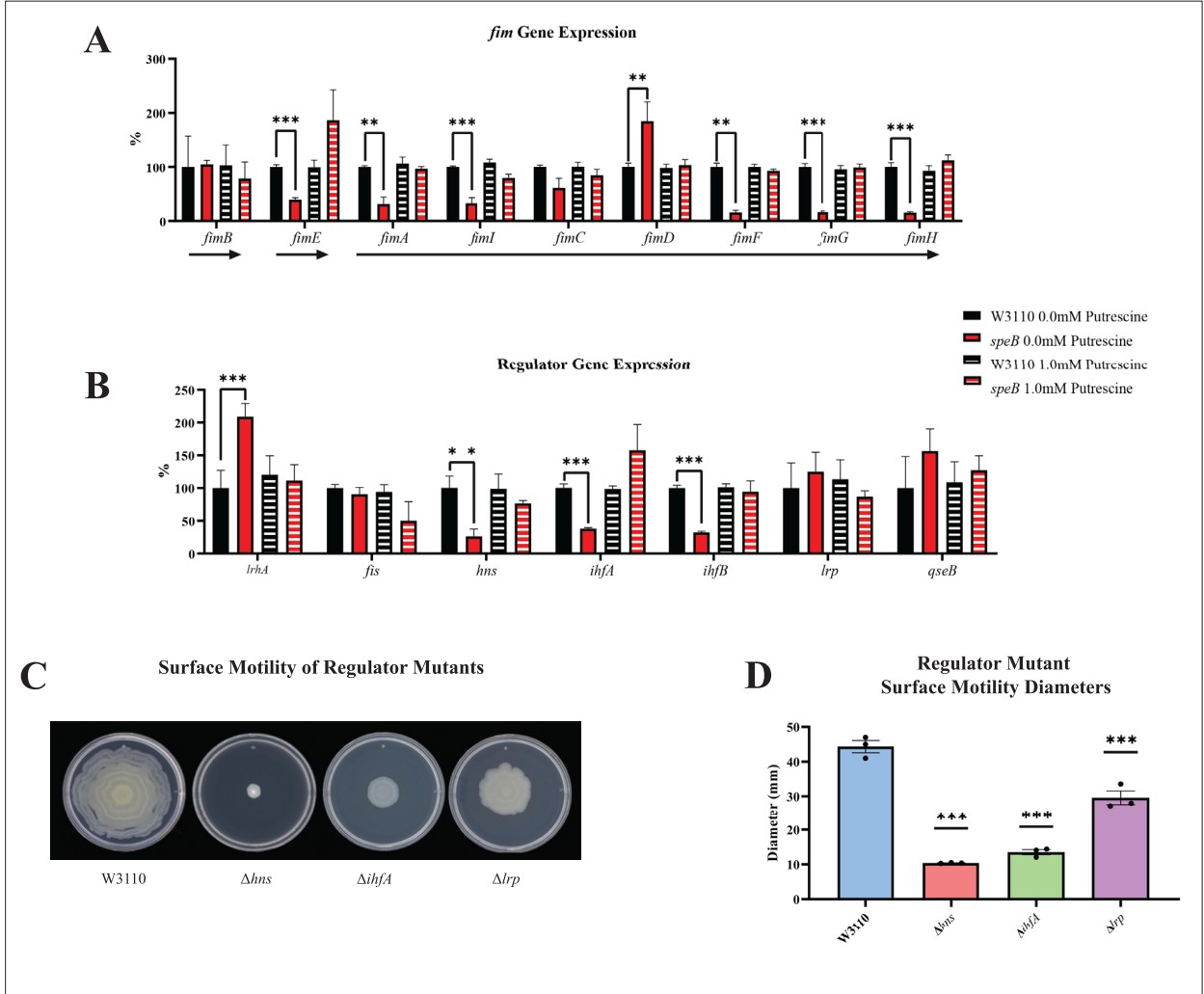

**Figure 10.** Transcriptomic sequencing of genes for the *fim* region and regulators that control *fim* gene expression. (**A**) Expression of the *fim* genes in the parental W3110 and the *speB* mutant with and without putrescine supplementation. Values were calculated by dividing the individual replicates' counts per million (CPM) value for each gene by the mean CPM of that gene in W3110 grown without putrescine and multiplying by 100 to yield the percent expression. Arrows below the genes signify the known operons: *fimB* and *fimE* belong to single gene operons, while *fimAICDFGH* belongs to one operon. One-way ANOVA was used to determine significance using Dunnett hypothesis testing. **p<0.01; ***p<0.001. (**B**) Expression of some regulators known to affect *fim* gene expression. Values were calculated as described in (**A**). ***p<0.001. (**C**) Surface motility of W3110 and three regulator mutants (Δ*hns*, Δ*ihfA*, and Δ*lrp*). A gene found to be significantly different by this transcriptomic analysis (*hns*) was confirmed to be important in surface motility. In this figure the Δ*speB* mutant was J15. (**D**) Diameter of surface movement of regulatory mutants after 36 hr. Error bars represent standard deviations for three independent replicates. Statistical analysis was performed using the Dunnett test of significance: *p<0.05; **p<0.01; ***p<0.001. All assays were performed in triplicate.

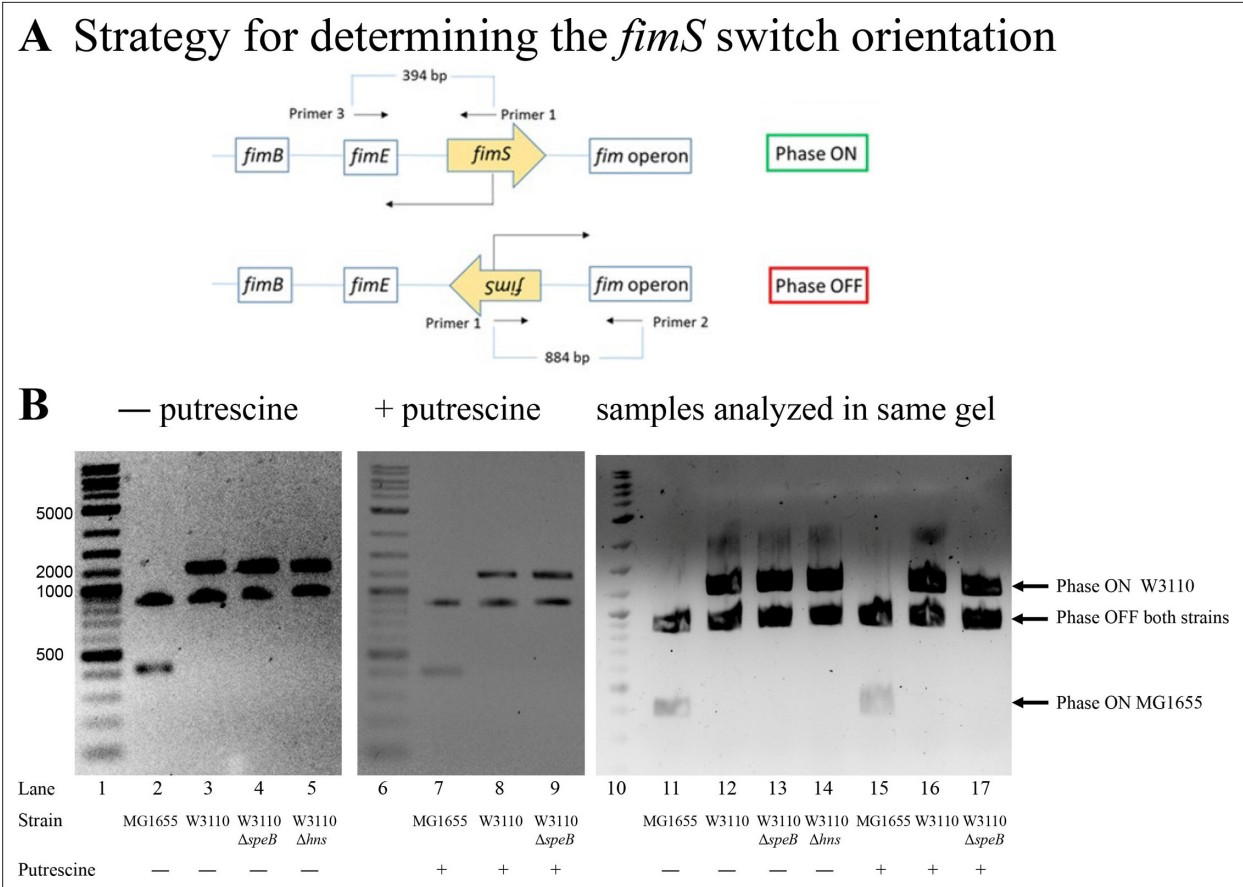

**Figure 11.** Phase variation in parental W3110, W3110 Δ*speB*, W3110 Δ*hns*, and MG1655. (**A**) The diagram shows the genes for the FimB and FimE recombinases, the invertible *fimS* region which contains the promoter for the *fim* operon, and the *fim* operon which codes for the proteins of the type 1 pilus. Primer pairs 1–2 and 1–3 detect the *fimS* region in the phase OFF and ON orientations, respectively. The DNA sizes for phases OFF and ON are 884 and 394, respectively, for wild-type strains of *E. coli*, such as MG1655. Our lab strain of W3110 has an *IS1* element insertion in *fimE* which increases the size of the amplified DNA fragment. MG1655 was analyzed as a control. Primer 1 is 5'-CCGCGATGCTTTCCTCTATG-3'; primer 2 is 5'-TAATGACG CCCTGAAATTGC-3'; and primer 3 is 5'-TGCTAACTGGAAAGGCGCTG-3' (shown schematically). (**B**) Deletion of either *speB* or *hns* had no effect on *fimS* orientation. A possible explanation for the loss of pili or PDSM in the *speB* or *hns* mutants is locking the *fimS* switch in phase OFF. However, loss of either *speB* or *hns* had no effect on *fimS* orientation in W3110 (lanes 2–4), and putrescine did not alter *fimS* orientation of the W3110 Δ*speB* mutant (lanes 6 and 7). We conclude that loss of *speB* in W3110 did not phase-lock *fimS* in phase OFF. Also note that W3110, which has an insertion in *fimE*, is not locked in phase ON.

intracellular putrescine (*Figure 11*). The *speB* mutant also had more transcripts for magnesium and phosphate transport genes. The elevated transcripts for magnesium transport genes could compensate for low putrescine to the extent that magnesium can directly replace putrescine, e.g., binding to nucleotides and RNA (*Miyamoto et al., 1993*). Two recent studies observed an inverse correlation between intracellular magnesium and polyamines in *Salmonella* and proposed that either magnesium or the polyamines maintained an overall divalent cation homeostasis (*Iwadate et al., 2023*; *Duprey and Groisman, 2020*). The rationale for an increase in anionic phosphate assimilation is less clear, although the major phosphate permease transports metal-phosphates, including magnesium-phosphate (*van Veen et al., 1994*), which could help to compensate for low putrescine. Regardless of the rationale, the substantial changes in gene expression due to low putrescine suggest multiple compensatory mechanisms.

A major observation of the RNAseq analysis is that loss of *speB* lowered transcripts for many genes of energy metabolism and all genes of iron acquisition. This gene expression pattern suggests that low putrescine diverts metabolism away from energy generation and toward putrescine synthesis, i.e., a prioritization of putrescine synthesis over aerobic energy metabolism. Oxidative energy metabolism and putrescine synthesis compete for α-ketoglutarate and the metabolic diversion could also be a

**Table 2.** Differentially expressed genes that are proposed to contribute to putrescine homeostasis.
A positive number means more transcripts in the Δ*speB* (lower putrescine) strain.

| Gene (function) | log$_2$FC | FDR |
| --- | --- | --- |
| **Polyamine synthesis** | | |
| *speA* (putrescine) | 0.99 | 0.003 |
| *speC* (putrescine) | 1.43 | 3E-4 |
| *speD* (spermidine) | −0.83 | 0.01 |
| *speF* (putrescine) | 1.05 | 0.007 |
| *rpmH* (inhibitor of SpeA and SpeC) | −2.77 | 2E-4 |
| *rpsT* (inhibitor of SpeA and SpeC) | −1.92 | 8E-4 |
| | | |
| **Polyamine transport** | | |
| *potABCD** (spermidine) | −2.17 | 3E-4 |
| *potE* (putrescine) | 2.07 | 2E-4 |
| *potFGHI** (putrescine) | 1.41 | 4E-4 |
| *plaP* (putrescine) | 1.11 | 0.005 |
| *sapBCDF** (putrescine export) | −1.18 | 8E-4 |
| | | |
| **Arginine synthesis** | | |
| *argA_2* (synthesis) | 1.77 | 0.001 |
| *argB* (synthesis) | 1.87 | 0.002 |
| *argC* (synthesis) | 2.86 | 0.005 |
| *argD* (synthesis) | 1.41 | 0.001 |
| *argF* (synthesis) | 1.63 | 0.01 |
| *argG* (synthesis) | 2.32 | 0.002 |
| *argH* (synthesis) | 0.97 | 0.001 |
| *argI* (synthesis) | 1.95 | 0.018 |
| *argR* (repressor of arginine regulon) | −1.35 | 4E-4 |
| **Glutamate generation** | | |
| *glsA* (glutamine degradation) | 2.16 | 5E-4 |
| *glnG* (regulation of ammonia assimilation) | −1.59 | 0.022 |
| | | |
| **TCA cycle/electron transport** | | |
| *acnA* (TCA cycle) | −1.04 | 8E-4 |
| *acnB* (TCA cycle) | −1.96 | 2E-4 |
| *fumA* (TCA cycle) | −1.50 | 6E-4 |
| *fumB* (TCA cycle) | −1.49 | 0.003 |
| *gltA* (TCA cycle) | −1.64 | 2E-4 |
| *icdA* (TCA cycle) | −1.32 | 0.002 |
| *sdhCDAB-sucABCD** (TCA cycle) | −3.88 | 5E-5 |

*Table 2 continued on next page*

*Table 2 continued*

| Gene (function) | log$_2$FC | FDR |
| --- | --- | --- |
| *nuo* operon* (electron transport) | –0.82 | 0.004 |
| *menFDHBCE** (electron transport) | –0.8 | 0.005 |
| *ubiEJB** (electron transport) | –0.93 | 0.004 |
| *cyoABCD** (electron transport) | –1.41 | 9E-4 |
| *hemCD** (heme) | –2.05 | 2E-4 |
| | | |
| **Iron transport** | | |
| *exbBD** (transport of all iron chelates) | –2.33 | 4E-5 |
| *tonB* (transport of all iron chelates) | –1.9 | 4E-5 |
| *entCEBA** (enterochelin iron) | –5.2 | 7E-6 |
| *fecABCDE * (ferric citrate) | –4.4 | 3E-5 |
| *fecIR* (regulators, ferric citrate) | –4.4 | 6E-6 |
| *feoABC** (ferrous iron) | –3.4 | 7E-4 |
| *fur* (regulator, iron assimilation) | –0.40 | 0.11 |
| | | |
| **Magnesium and phosphate transport** | | |
| *mgtA* (magnesium) | 5.90 | 5E-6 |
| *phoQ* (regulator, magnesium) | 0.79 | 0.009 |
| *pstSCAB** (phosphate) | 3.61 | 0.01 |
| *phoBR** (regulator, phosphate assimilation) | 2.80 | 0.02 |

*Values for transcripts of the first gene of the operon are given. Results for other genes of the operon are in **Supplementary file 2**.

potential compensatory mechanism for low putrescine. Another way of looking at this relationship is that intracellular putrescine positively correlates with oxidative energy metabolism. Deletion of *ptsH*, *ptsG*, *gltA*, *sdhA*, and *sucC* have been shown to impair W3110 PDSM which suggests that a major, possibly the primary, effect of low putrescine is reduced energy generation.

In contrast to our results, a thorough and well-designed study showed that *E. coli* PDSM requires spermidine instead of putrescine (**Kurihara et al., 2009**). The differences between the studies were strain backgrounds, media (glucose-LB vs glucose-tryptone), and incubation temperature (33° vs 37°). We found that PDSM results for 37° incubations were highly variable and resulted in numerous fast-moving variants. Since growth at the higher temperature is faster, and faster growth is associated with higher putrescine (**Tweeddale et al., 1998**), we propose that growth at 37° results in an inhibitory putrescine concentration. In this case, spermidine would be stimulatory because of its known inhibition of the putrescine biosynthetic enzymes SpeA and SpeC (**Applebaum et al., 1977**; **Wu and Morris, 1973**) and subsequent reduction of intracellular putrescine to an optimal concentration. Regardless of the explanation, strain differences are not unexpected because of decades of strain passage for the strains employed with resulting laboratory evolution that can affect any of the numerous factors that control the intracellular putrescine concentration.

The transcript changes observed in our study were different from those in the only other systematic study of putrescine's effect on *E. coli* gene expression (**Yoshida et al., 2004**). Both studies had almost identical experimental designs: 1.0- or 1.14 mM putrescine was added to a *speB* or *speB speC* mutant, respectively. Differences in strains, growth media (minimal media vs tryptone-containing media), and growth temperature (33° vs 37°) could account for the discrepancy in results. The most crucial difference could be the growth media: Igarashi and colleagues grew bacteria in a minimal medium, while we grew bacteria in an amino acid-containing medium that was required for surface motility. We propose

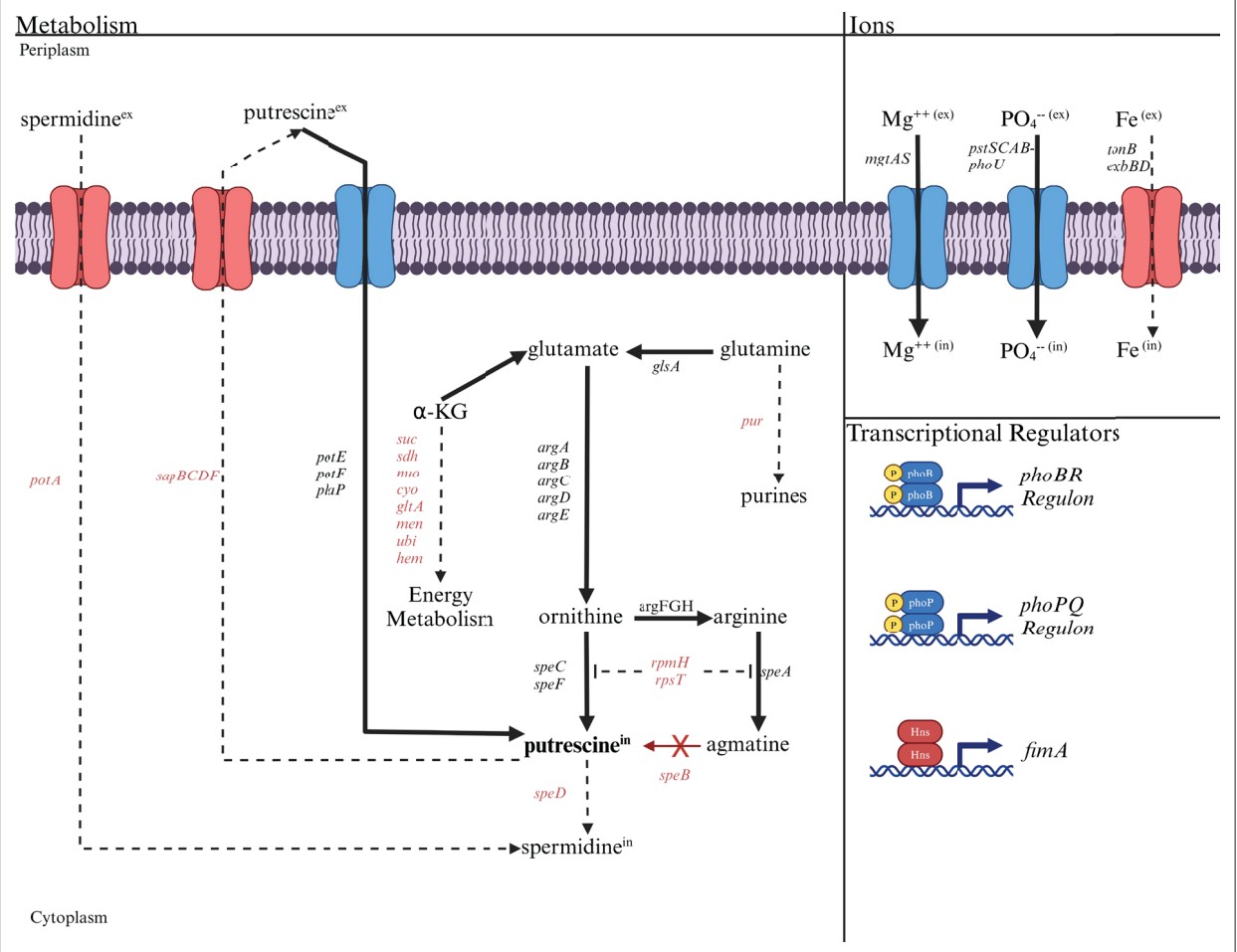

**Figure 12.** The deduced diversion of metabolism in a *speB* mutant away from energy metabolism toward putrescine synthesis compared to the parental strain. The effects of low putrescine are shown. Genes and processes (transport and transcriptional regulators) in red have fewer transcripts, while those in black or blue have more. The dashed lines represent the proposed reduction in metabolic flux because of fewer transcripts from genes coding for the enzymes involved. Operons are shown except for the larger operons or regulons. For example, *pur* is meant to represent the unlinked genes that code for enzymes of purine synthesis. *Table 2* or *Supplementary file 2* should be consulted for specific genes and the quantitative change in transcripts.

that the slower growth in minimal medium, possibly correlated with higher guanosine tetraphosphate which is associated with slower growth, counters and obscures many growth-stimulatory putrescine effects. In contrast, the growth stimulation from the amino acids in our medium would not counter putrescine effects and would essentially allow the detection of effects caused solely by changes in putrescine concentrations. Regardless of the explanation, common results can be interpreted as core responses to putrescine limitation. Both studies observed that low putrescine increased transcripts for genes of arginine synthesis and magnesium transport, and reduced transcripts for the *sdhCDAB-sucABCD* operon which codes for three TCA cycle enzymes. The link between putrescine and arginine synthesis is probably mediated by the ArgR repressor because *argR* transcripts are 2.5-fold lower (FDR = 0.0004) in the *speB* mutant (*Supplementary file 2*).

Several results implicate putrescine in *E. coli* virulence during urinary tract infections. First, pili-dependent binding to the urothelium is an essential step for uropathogenic *E. coli* (*Flores-Mireles et al., 2015*). Second, putrescine is present in urine from infected, but not healthy, individuals (*Puebla-Barragan et al., 2020*; *Satink et al., 1989*). Third, urinary putrescine can result from uropathogenic *E. coli* growth in urine (*Satink et al., 1989*) and urothelial pathophysiology (*Keay et al., 2014*). Putrescine's possible role in virulence is seemingly contradicted by the observation that a *speB* mutant outcompetes the parental CFT073 in the bladder of mice, i.e., less putrescine synthesis enhanced virulence (*Alteri et al., 2009*). A majority of uropathogenic *E. coli* is associated with phylogenetic

group B2—including CFT073, whereas most *E. coli* lab strains, such as W3110, are group A. Group B2 strains lack genes for the major putrescine catabolic pathway and have significantly higher transcripts for genes coding for SpeA and SpeB (*Hogins et al., 2023a*). Both observations suggest that the cytoplasmic putrescine concentration is higher in group B2 strains. We propose that the putrescine concentration in group B2 strains is normally in the inhibitory range for pili synthesis, and that loss of SpeB reduces putrescine to a level that stimulates pili synthesis. Regardless of the explanation, putrescine control of pili synthesis and other processes in uropathogenic *E. coli* are worth examination.

# Materials and methods

**Key resources table**

| Reagent type (species) or resource | Designation | Source or reference | Identifiers | Additional information |
|---|---|---|---|---|
| Gene (*Escherichia coli*) | *speB* | PMID:16738554 | | P1 transduction construction |
| Gene (*E. coli*) | various, see strain list | PMID:16738554 | | P1 transduction construction |
| Strain, strain background (*E. coli*) | W3110 | other | | Lab strain; history provided in Methods |
| Genetic reagent (*E. coli* transducing virus) | P1 vir | other | | Lab strain |
| Antibody | anti-*E. coli* RpoD (rabbit polyclonal) | Cusabio | Cat #: CSB-PA360419XA01ENV RRID:AB_3678626 | 1:10,000 |
| Antibody | anti-*E. coli* FimA (rabbit polyclonal) | Cusabio | Cat #: CSB-PA361210ZA01ENV RRID:AB_3678627 | 1:10,000 |
| Antibody | Goat anti-rabbit IgG (H+L) HRP conjugated | Cusabio | Cat #: CSB-PA489724 RRID:AB_3678628 | 1:10,000 |
| Sequence-based reagent | FimA forward | This paper | PCR primers | ATGGTGGGACCGTTCACTTT |
| Sequence-based reagent | FimA reverse | This paper | PCR primers | GGCAACAGCGGCTTTAGATG |
| Sequence-based reagent | RpoD forward | This paper | PCR primers | TCGTGTTGAAGCAGAAGAAGCG |
| Sequence-based reagent | RpoD reverse | This paper | PCR primers | TCGTCATCGCCATCTTCTTCG |
| Sequence-based reagent | Phase variation test primer 1 | This paper | PCR primers | CCGCGATGCTTTCCTCTATG |
| Sequence-based reagent | Phase variation test primer 2 | This paper | PCR primers | TAATGACGCCCTGAAATTGC |
| Sequence-based reagent | Phase variation test primer 3 | This paper | PCR primers | TGCTAACTGGAAAGGCGCTG |
| Commercial assay or kit | TMB substrate kit | ThermoFisher | Cat #: 34021 | |
| Commercial assay or kit | Ligation sequencing DNA V14 | Oxford Nanopore Technologies | Cat #: SQK-LSK114 | |
| Commercial assay or kit | Rneasy Mini Kit | Qiagen | Cat #: 74104 | |

*Continued on next page*

*Continued*

| Reagent type (species) or resource | Designation | Source or reference | Identifiers | Additional information |
|---|---|---|---|---|
| Commercial assay or kit | RiboMinus Transcriptome Isolation Kit, bacteria | ThermoFisher | Cat #: K155004 | |
| Commercial assay or kit | RiboCop rRNA Depletion Kits for Bacteria | Lexogen | Cat #: 126.24 | |
| Commercial assay or kit | LunaScript RT Super | NEB | Cat #: E3010 | |
| Commercial assay or kit | PowerUp STBR Green Master Mix for qPCR | ThermoFisher | Cat #: A25777 | |
| Chemical compound, drug | Eiken Agar | Eiken Chemical Co., Ltd, Tokyo, Japan | Cat #: E-MJ00 | |
| Software, algorithm | bcl2fastq (v 4.2.4) | Illumina | RRID:SCR_015058 | |
| Software, algorithm | Guppy basecaller (v 6.5.7) | Oxford Nanopore Technologies | RRID:SCR_023196 | |
| Software, algorithm | Porechop (v 0.2.4) | Oxford Nanopore Technologies | RRID:SCR_016967 | |
| Software, algorithm | Flye (v 2.9.2) | PMID:27956617 | RRID:SCR_017016 | |
| Software, algorithm | Pilon (v 1.24) | https://doi.org.10.1371/journal.pone.0112963 | RRID:SCR_014731 | |
| Software, algorithm | Circulator (v 1.5.5) | PMID:26714481 | | |
| Software, algorithm | Prokka (v 1.14.6) | https://doi.org.10.1093/bioinformatics/btu153 | RRID:SCR_014732 | |
| Software, algorithm | QUAST (v 5.2.0) | PMID:23422339 | RRID:SCR_001228 | |
| Software, algorithm | CLC Genomics Workbench | Qiagen | RRID:SCR_011853 | |
| Software, algorithm | edgeR | PMID:19910308 | RRID:SCR_012802 | |

## Strains

All strains used for growth rate determinations and motility assays were derivatives of *E. coli* K-12 strain W3110 and are listed in *Table 3*. Great variations exist in standard lab strains, including W3110, from different labs. We initially tested nine strains derived from *E. coli* K-12, mostly from the Coli Genetic Stock Center (*Ambagaspitiye et al., 2019*), and chose our lab strain because of relative genetic stability and more quantitatively reproducible results. Our strain originated from the lab of Jon Beckwith in the 1960s via the lab of Boris Magasanik where it had been stored in a room temperature stab until the mid-1980s, at which time it was revived and frozen at –80 °C. To construct the mutant strains, the altered allele was obtained from the Keio collection in which the gene of interest had been deleted and replaced with an antibiotic-resistance gene (*Baba et al., 2006*). The marked deletion allele was transferred into W3110 by P1 transduction (*Miller, 1972*). The antibiotic gene was removed as described which generated an in-frame deletion (*Datsenko and Wanner, 2000*).

**Table 3.** Strains.

| Strain name | Genotype | Reference |
|---|---|---|
| BLS77 | W3110 Δ*puuR::cat* | *Schneider and Reitzer, 2012* |
| BLS80 | W3110 Δ*puuA::cat* | *Schneider and Reitzer, 2012* |
| BLS88 | W3110 Δ*patA* Δ*puuA* | *Schneider and Reitzer, 2012* |
| CP2 | W3110 Δ*patA* | *Schneider and Reitzer, 2012* |
| IM26 | W3110 Δ*speB::kan fliC-lacZ* | This study |
| IM27 | W3110 Δ*speE::kan fliC-lacZ* | This study |
| IM28 | W3110 Δ*speF::cat fliC-lacZ* | This study |
| IM29 | W3110 Δ*speC::kan fliC-lacZ* | This study |
| IM34 | W3110 Δ*speC* Δ*speF::cat fliC-lacZ* | This study |
| IM60 | W3110 Δ*cadA::kan fliC-lacZ* | This study |
| IM61 | W3110 Δ*speA::kan* | This study |
| IM62 | W3110 Δ*patA* Δ*puuA* Δ*speE::kan* | This study |
| IM63 | W3110 Δ*potE::kan* | This study |
| IM64 | W3110 Δ*potF::kan* | This study |
| IM65 | W3110 Δ*plaP::kan* | This study |
| IM66 | W3110 Δ*puuP::kan* | This study |
| J15 | W3110 Δ*speB* Δ*fliC::kan* | This study |
| SA1 | W3110 Δ*fliC::kan* | This study |
| SA2 | W3110 Δ*fimA::kan* | This study |
| SA3 | W3110 Δ*fliC* Δ*fimA* | This study |
| SA4 | W3110 Δ*hns::kan* | This study |
| SA5 | W3110 Δ*ihfA::kan* | This study |
| SA6 | W3110 Δ*lrp::kan* | This study |
| W3110-LR referred to as W3110 | *lacI^q lacL8* | Lab strain |

## Media and growth conditions

For P1 transductions and plasmid transformation experiments, cells were grown in standard LB liquid medium (1% tryptone, 1% NaCl, and 0.5% yeast extract, pH 7.0) at 37°C. Antibiotics were used for selection at concentrations of 25 µg/mL (both chloramphenicol and kanamycin). For growth analysis, bacteria were grown in a liquid motility medium (0.5% glucose, 1% tryptone, and 0.25% NaCl) which we refer to as GT medium. Starter cultures (typically 6–12 hr incubation) were grown in GT medium, harvested by centrifugation, washed twice with phosphate-buffered saline, and re-suspended in GT medium before inoculation. The cells were then grown at 37 °C in aerobic conditions (shaking at 240 rpm) and the turbidity was measured every 30 min. Cell growth was measured in Klett units using a Scienceware Klett colorimeter with a KS-54 filter. 100 Klett units represent an $OD_{600}$ value of about 0.7.

## Surface motility assay

For a standard surface motility assay, single colonies from fresh plates (streaked out from frozen stocks a day before) were inoculated in GT medium for 6 hr at 37 °C in aerobic conditions (shaking at 240 rpm). 30 ml of autoclaved GT medium with 0.45% agar (Eiken, Tokyo, Japan) was poured into a sterile polystyrene petri dish (100 mm × 15 mm) and allowed to solidify at room temperature for approximately 6 hr. Then, 1 µL of the pre-motility growth medium was inoculated at the center of the agar plate and incubated at 33°C for 36 hr. Each experiment was performed in triplicate and pictures

were taken after 36 hr. Surface motility was extremely sensitive to humidity. Opening the incubator before 36 hr resulted in movement cessation. The surface motility assay was performed at 33 °C because results from incubations at 37°C were highly variable and the cultures more frequently generated genetically stable fast-moving variants.

## Swim assay

The media and culturing for swimming motility are identical to that for surface motility, except that the plates contained 0.25% agar and were solidified at room temperature for about 1 hr before inoculation. Then, 1 µL of the culture was stabbed in the center of this swim agar plate and incubated at 33°C for 20 hr. Each experiment was performed in triplicate.

## Transmission electron microscopy

Cells from surface motility plates were collected and fixed with 2.5% glutaraldehyde. Bacteria were absorbed onto Foamvar carbon-coated copper grids for 1 min. Grids were washed with distilled water and stained with 1% phosphotungstic acid for 30 s. Washed and stained grids were dried at 37 °C for 10 min. Samples were viewed on a JEOL 1200 EX transmission electron microscope at the University of Texas Southwestern Medical Center (*Figure 2*) and the University of Texas at Dallas (*Figure 8*).

## ELISA analysis

Cells were grown overnight in 5 mL GT media followed by a 2 hr growth in GT media with 0.0, 0.1, 1.0, or 4.0 mM putrescine. Cells were then lysed using a 24-gauge needle. The cell lysate was diluted to a concentration of 1 mg/mL protein (based on A280) in pH 9.6 coating buffer (3 g $Na_2CO_3$, 6 g $NaHCO_3$, 1000 mL distilled water) and then coated onto the walls of a 96-well plate by incubating overnight at 4 °C. The wells were rinsed three times with PBS and further blocked using coating buffer with 1% bovine albumin overnight at 4 °C. The following day, wells were rinsed three times with PBS, and primary antibodies to FimA or RpoD were added following supplier directions. After a 2 hr room temperature incubation, secondary antibodies conjugated to HRP were added and incubated for a further 2 hr. HRP was activated using the TMB substrate kit (Thermo-34021) following the provided protocol. Expression was read on a BioTek plate reader based on the provided kit protocol. Data was blanked to wells only treated with bovine albumin and then normalized to RpoD. Data was generated from technical and biological triplicates.

## DNA isolation and genome assembly and annotation for W3110

DNA was isolated based on previously established protocols (*Hogins et al., 2023b*). Short reads were sequenced at the SeqCenter (Pittsburgh, Pennsylvania) using Illumina tagmentation-based and PCR-based DNA prep and custom IDT indices targeting inserts of 280 bp without further fragmentation or size selection steps. The Illumina NovaSeq X Plus sequencer was run producing 2×151 paired-end reads. Demultiplexing, quality control, and adapter trimmer were performed with bcl-convert (v4.2.4). Total Illumina Reads (R1+R2): 4059078 with 553087045 bps >Q30.

Long reads were prepared using the PCR-free Oxford Nanopore Technologies ligation sequencing kit with the NEBNext companion module. No fragmentation or size selection was performed. Long-read libraries were performed using the R10.4.1 flow cell. The 400bps sequencing mode with a minimum read length of 200 bp was selected. Guppy (v6.5.7) was used at the super-accurate base calling, demultiplexing, and adapter removal. Total long reads: 53773 with 89.607% of bps >Q20.

To generate the completed genome, porechop (v0.2.4) was used to trim residual adapter sequences from long reads. Flye (v2.9.2) was used to generate the de novo assembly under the nano-hq model. Reads longer than the estimated $N_{50}$ based on a genome size of 6Mbp initiated the assembly. Subsequent polishing using the short read data was performed using Pilon (v1.24) (RRID:SCR_014731) under default parameters. Long-read contigs with an average short-read coverage of 15 X or less were removed from the assembly. The assembled contig was confirmed to be circular via circulator (v1.5.5). Annotation was performed using prokka (v 1.14.6). Finally, statistics were recorded using QUAST (v5.2.0). The final genome contained 1 contig of 4750347 bp with a sequencing depth of 123.92 x. The $N_{50}$ was 4750347. This genome can be accessed via the accession number CP165600 or via the BioProject PRJNA1142534 via NCBI. The completed genome has an estimated average nucleotide identity of 99.9694 to the W3110 genome deposited to NCBI (genome assembly ASM1024v1).

## RNA isolation and quality control

Cells were grown for 2 hr in 1 mL GT media. 60 μL were then added to 1 mL of fresh GT, with or without 1 mM putrescine, and grown for another 2 hr. After growth, the cells were centrifuged, and frozen at –80 °C. The isolation and analysis protocol has been previously published (*Hogins et al., 2023a*). Cell pellets were thawed, resuspended in 0.7 mL of buffer RLT (Qiagen RNeasy kit), and mechanically lysed using a bead beater (FastPrep-24 Classic from MP Biomedical (RRID:SCR_013308)) set at the highest setting for three 45 s cycles with a 5 min rest period on ice between cycles. Cell lysates were used to isolate RNA using a Qiagen RNeasy mini kit. RNA was quantified, DNase treated, and re-quantified on a nanodrop. RNA that passed the first quality check was analyzed on RNA-IQ (Qubit). RNA with an IQ higher than 7.5 was used. RNA that passed both checks was run on an agarose gel to check for RNA integrity. RT-qPCR was used to ensure there was no DNA contamination. RNA libraries that passed all these quality checks were submitted to the genomics core facility at The University of Texas at Dallas for RNA sequencing. The core performed rRNA removal (RiboMinus Transcriptome Isolation Kit or RiboCop bacterial rRNA depletion—Lexogen), library preparation (Stranded Total RNA Prep—Illumina), and single-end Illumina sequencing.

## Reverse transcription quantitative PCR

RNA was isolated as described above. One microgram of RNA was reverse transcribed using Luna-Script RT Super Mix (NEB E3010) and another microgram was subjected to the same reaction as the reverse transcribed RNA, except without reverse transcriptase (negative control) following manufacturer instructions. The cDNA was then diluted to 10 ng/μL for qPCR. PCR reactions were composed of 10 ng cDNA, 8 μL of nuclease-free water, 10 nM primers (rpoD: Forward: TCGTGTTGAAGCAGAA GAAGCG; Reverse: TCGTCATCGCCATCTTCTTCG) (fimA: Forward: ATGGTGGGACCGTTCACTTT ; Reverse: GGCAACAGCGGCTTTAGATG), and PowerUp SYBR Green 2 X master mix for qPCR (ThermoFisher A25777) in a 20 μL reaction. A Quantstudio 3 qPCR machine was used to generate critical threshold (Ct) values. The Ct values were then analyzed using the $2^{-\Delta\Delta C_T}$ method (*Livak and Schmittgen, 2001*) using *rpoD* as a library control.

## RNAseq analysis

Transcripts were aligned to the W3110 genome and CDS (CP165600) available on NCBI using the CLC genomics workbench (Qiagen version 22.00) for RNA-seq analysis. The gene expression counts generated as this output were used to perform differentially expressed gene (DEG) analysis using EdgeR (version 3.38.4) (RRID:SCR_012802) to analyze read counts (*McCarthy et al., 2012*; *Robinson et al., 2010*). DEGs had an adjusted p-value of less than 0.05.

## Material availability statement

All strains are available upon request. All mutant alleles are readily accessible via the KEIO mutant collection (*Baba et al., 2006*).

# Acknowledgements

This work was funded through private donations. PEZ received endowment support from the Cain Foundation through the Felecia and John Cain Distinguished Chair in Women's Health in honor of Philippe E Zimmern, MD. The authors declare no potential conflicts of interest.

The authors (JH, SH, GV, and LR) would like to thank Prof. Kelli Palmer of The University of Texas at Dallas for the use of her space, supplies, and tissue homogenizer. JH would like to further thank Prof. Palmer for the use of her CLC genomics program. The authors would like to thank the Genome Center at The University of Texas at Dallas for the services to support our research. The authors would like to thank the Olympus Discovery Center/Imaging Core facility at UT Dallas for providing equipment and support (*Figure 8*), and the UT Southwestern Medical Center for electron microscopy (*Figure 2*).

# Additional information

## Funding

| Funder | Grant reference number | Author |
| --- | --- | --- |
| Cain Foundation | The Felecia and John Cain Distinguished Chair in Women's Health, in Honor of Philippe Zimmern, M.D. | Philippe E Zimmern |

This work was funded through private donations. PEZ received endowment support from the Cain Foundation through the Felecia and John Cain Distinguished Chair in Women's Health in honor of Philippe E Zimmern, MD. The funders had no role in study design, data collection and interpretation, or the decision to submit the work for publication.

## Author contributions

Iti Mehta, Formal analysis, Validation, Investigation, Visualization, Writing – original draft; Jacob B Hogins, Data curation, Formal analysis, Supervision, Validation, Investigation, Visualization, Methodology, Writing – original draft, Project administration, Writing – review and editing; Sydney R Hall, Formal analysis, Investigation, Visualization, Methodology; Gabrielle Vragel, Formal analysis, Validation, Investigation, Visualization, Methodology; Sankalya Ambagaspitiye, Formal analysis, Validation, Investigation, Visualization; Philippe E Zimmern, Supervision, Funding acquisition, Writing – review and editing; Larry Reitzer, Conceptualization, Resources, Data curation, Formal analysis, Supervision, Funding acquisition, Validation, Investigation, Visualization, Methodology, Writing – original draft, Project administration, Writing – review and editing

## Author ORCIDs

Jacob B Hogins ⓘ https://orcid.org/0000-0001-6041-7332
Larry Reitzer ⓘ https://orcid.org/0000-0002-4406-6090

Reviewer #2 (Public review): https://doi.org/10.7554/eLife.102439.3.sa1
Reviewer #3 (Public review): https://doi.org/10.7554/eLife.102439.3.sa2
Author response https://doi.org/10.7554/eLife.102439.3.sa3

---

# Additional files

## Supplementary files

Supplementary file 1. Expression graphs comparing the logCPM of the transcriptomes of the average of the three replicates of the W3110 with and without 1 mM putrescine and the speB mutant with and without 1 mM putrescine. A regression line was calculated and the correlation between each set of transcriptomes was noted on the graph. Higher $R^2$ values indicate greater similarity between the transcriptomes.

Supplementary file 2. Differential gene expression analysis of W3110 and Δ*speB*. J15 refers to the Δ*speB* mutant. In the tab labeled 'W3110_0_vs_J15_1_ Putrescine' W3110 was grown without putrescine, and J15 was grown with 1.0 mM putrescine; positive fold change values refer to increases in J15 gene expression.

MDAR checklist

## Data availability

The raw RNA seq reads were deposited in GenBank: BioProject accession number PRJNA1126736. The genome was deposited in GenBank: BioProject accession number PRJNA1142534. The genome accession number is CP165600.

The following datasets were generated:

| Author(s) | Year | Dataset title | Dataset URL | Database and Identifier |
|---|---|---|---|---|
| Hogins J, Reitzer L | 2024 | Transcriptomic Sequencing of a speB mutant in *Escherichia coli* W3110 | https://www.ncbi.nlm.nih.gov/bioproject/?term=PRJNA1126736 | NCBI BioProject, PRJNA1126736 |
| Hogins J, Reitzer L | 2024 | *Escherichia coli* str. K-12 substr. W3110 Genome sequencing | https://www.ncbi.nlm.nih.gov/bioproject/?term=PRJNA1142534 | NCBI BioProject, PRJNA1142534 |
| Hogins J, Reitzer L | 2025 | Escherichia coli str. K-12 substr. W3110 chromosome, complete genome | https://www.ncbi.nlm.nih.gov/nuccore/CP165600 | NCBI Nucleotide, CP165600 |

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
