## [Editor Report · eLife Assessment]

This **valuable** study presents an interesting analysis of the role of the polyamine precursor putrescine in the pili-dependent surface motility of a laboratory strain of *Escherichia coli*. The overall data **convincingly** demonstrate a role in this case. This study presents interesting findings for those studying uropathogenic bacteria, and those studying bacterial polyamine function.

---

## [Referee Report · Reviewer #2 (Public review)]

Summary:

Mehta et al., in constructing *E. coli* strains unable to synthesize polyamines, noted that strains deficient in putrescine synthesis showed decreased movement on semisolid agar. They show that strains incapable of synthesizing putrescine have decreased expression of Type I pilin and, hence, decreased ability to perform pilin-dependent surface motility.

Strengths:

The authors characterize the specific polyamine pathways that are important for this phenomenon. RNAseq provides a detailed overview of gene expression in the strain lacking putrescine. They rule out potential effects of pilin phase variation on the phenotype. The data suggest homeostatic control of polyamine synthesis and metabolic changes in response to putrescine.

Weaknesses:

The authors do not, in the end, uncover the molecular details of pilin expression per se, but that would require significantly more analyses and data; the mechanisms of pilin regulation are complicated and still not completely understood.

---

## [Referee Report · Reviewer #3 (Public review)]

Summary:

This study by Mehta et al. describes the mechanisms behind the observation that putrescine biosynthesis mutants in *Escherichia coli* strain W3110 are affected in surface motility. The manuscript shows that the surface motility phenotype is dependent on Type I fimbriae and that putrescine levels affect the expression level of fimbriae. The results further suggest that without putrescine, the metabolism of the cell is shifted towards production of putrescine and away from energy metabolism.

Strengths:

The authors show the effect of putrescine on the regulation of type I fimbriae using various strategies (mutants, addition of exogenous, RNA seq, etc.). All experiments converge to the same conclusion that an optimal level of putrescine is needed.

Weakness:

The authors use one isolate of *E. coli* strain W3110, that contains an insertion in fimE which controls the expression of type I fimbriae. The insertion in fimE likely modifies the ratio of cells expressing fimbriae in the population, and it would be important to confirm the results in other isolates or other strains.

---

## [Author Response]

The following is the authors’ response to the original reviews.

**Alternate explanations for major conclusions**.

The major conclusions are (a) surface motility of W3110 requires pili which is not novel, (b) pili synthesis and pili-dependent surface motility require putrescine — 1 mM is optimal, and 4 mM is inhibitory, and (c) the existence of a putrescine homeostatic network that maintains intracellular putrescine that involves compensatory mechanisms for low putrescine, including diversion of energy generation toward putrescine synthesis.

Conclusion a: Reviewer 3 suggests that the mutant may have lost surface motility because of outer surface structures that actually mediate motility but are co-regulated with or depend on pili synthesis. The reviewer explicitly suggests flagella as the alternate appendage, although flagella and pili are reciprocally regulated. Most experiments were performed in a Δ*fliC* background, which lacks the major flagella subunit, in order to prevent the generation of fast-moving flagella-dependent variants. Furthermore, no other surface structure that could mediate surface motility is apparent in the electron microscope images. This observation does not definitively rule out this possibility, especially because of the large transcriptomic changes with low putrescine. Our explanation is the simplest.

Conclusion b, first comment: Reviewer 1 states that “it is not possible to conclude that the effects of gene deletions to biosynthetic, transport or catabolic genes on pili-dependent surface motility are due to changes in putrescine levels unless one takes it on faith that there must be changes to putrescine levels.” The comment ignores both the nutritional supplementation and the transcript changes that strongly suggest compensatory mechanisms for low putrescine. Why compensate if the putrescine concentration does not change? The reviewer then implicitly acknowledges changes in putrescine content: “it is important to know how much putrescine must be depleted in order to exert a physiological effect”.

Conclusion b, second comment: Reviewer 1 proposes that agmatine accumulation can account for some of the observed properties, but which property is not specified. With respect to motility, agmatine accumulation cannot account for motility defects because motility is impaired in (a) a *speA* mutant which cannot make agmatine and (b) a *speC speF* double mutant which should not accumulate agmatine. With respect to the transcriptomic results, even if high agmatine is the reason for some transcript changes, the results still suggest a putrescine homeostasis network.

Conclusion c: the reviewers made no comments on the RNAseq analysis or the interpretation of the existence of a homeostatic network.

**Additional experiments proposed.**

Complementation. Reviewers 1 and 3 suggested complementation experiments, but the latter states that nutritional supplementation strengthens our arguments. The most relevant complementation is with *speB*. We tried complementation and found that our control plasmid inhibited motility by increasing the lag time before movement commenced. A plasmid with *speB* did stimulate motility relative to the control plasmid, but movement with the *speB* plasmid took 4 days, while wild-type movement took 1.5 days. We think that interpretation of this result is ambiguous. We did not systematically search for plasmids that had no effect on motility.

The purpose of complementation is to determine whether a second-site mutation is the actual cause of the motility defect. In this case, the artifact is that an alteration in polyamine metabolism is not the cause of the defect. However, external putrescine reverses the effects on motility and pili synthesis in the *speB* mutant. This result is inconsistent with a second-site mutation. Still, we agree that complementation is important, and because of our difficulties, we tested numerous mutants with defects in polyamine metabolism. The results present an interpretable and coherent pattern. For example, if putrescine is not the regulator, then mutants in putrescine transport and catabolism should have had no effect. Every single mutant is consistent with a role in movement and pili synthesis. The simplest explanation is that putrescine affects movement and pili synthesis.

Phase variation. Reviewer 2 noted that we did not discuss phase variation. The comment came from the observation that the *speB* mutant had fewer *fimB* transcripts which could explain the loss of motility. The reviewer also suggested a simple experiment, which we performed and found that putrescine does not control phase variation. We present those results in the supplemental material. Our discussion of this topic includes a major qualification.

Testing of additional strains. Published results from another lab showed that surface motility of MG1655 requires spermidine instead of putrescine (PMID 19493013 and 21266585). MG1655 and the W3110 that we used in our study are *E. coli* K-12 derivatives and phylogenetic group A. Any number of changes in enzymes that affect intracellular putrescine concentration could result in different responses to putrescine. We are currently studying pili synthesis and motility in other strains. While that study is incomplete, loss of *speB* in a strain of phylogenetic group D eliminates no surface motility. This work was intended as our initial analysis and the focus was on a single strain.

Measuring intracellular polyamines. We felt that we had provided sufficient evidence to conclude that putrescine controls pili synthesis and putrescine concentrations are lower in the *speB* mutant: the nutritional supplementation, the lower levels of transcripts for putrescine catabolic enzymes which require putrescine for their expression strongly suggest lower putrescine in a mutant lacking a putrescine biosynthesis gene, and a transcriptomic analysis that found the *speB* mutant had transcript changes to compensate for low putrescine. We understand the importance of measuring intracellular polyamines. We are currently examining the quantitative relationship between intracellular polyamines and pili synthesis in multiple strains which respond differently to loss of *speB*.

**Recommendations for the authors:**

**Reviewer #1 (Recommendations for the authors):**
The authors should measure putrescine, agmatine, cadaverine, and spermidine levels in their gene deletion strains.

Polyamine concentration measurements will be part of a separate study on polyamine control of pili synthesis of a uropathogenic strain. A comparison is essential, and the results from W3110 will be part of that study.

**Reviewer #2 (Recommendations for the authors):**
(1) Line 28. Your statements about urinary tract infections are pure speculation. They are fine for the discussion, but should not be in the abstract.

The abstract from line 27 on has been reworked. The comment of the reviewer is fair.

(2) Line 65. Do we need this discussion about the various strains? If you keep it, you should point out that they were all W3110 strains. But you could just say that you confirmed that your background strain can do PDSM (since you are also not showing any data for the other isolates). Discussing the various strains implies that you are not confident in your strain and raises the question of why you didn't use a sequenced wt MG1655, or something like that.

This section has been reworked. Our strain of W3110 has an insertion in fimB which is relevant for movement but does not affect our results. The insertion limits our conclusions about phase variation. We want to point out that strains variations are large. We also sequenced our strain of W3110.

(3) Related. You occasionally use "W3110-LR" to designate the wild type. You use this or not, but be consistent throughout the text.

Fixed

(4) Line 99. Does eLife allow "data not shown"?(5) Line 119. As you note, the phenotype of the puuA patA double mutant is exactly the opposite of what one would expect. Although you provide additional evidence that high levels also inhibit motility, complementing the double mutant would provide confidence that the strain is correct.

We rapidly ran into issues with complementation which are discussed in public responses to reviewer comments.

(6) Figure 6C. Either you need to quantify these data or you need a better picture.

The files were corrupted. It was repeated several time, but we lost the other data.

(7) Figure 7. Label panels A and B to indicate that these strains are speB. Also, you need to switch panels C and D to match the order of discussion in the manuscript.

Done

(8) Line 134. Is there a statistically significant difference in the ELISA between 1 and 4 mM? You need to say one way or the other.

No statistical significance and this has been added to the paper

(9) Figure 10C. You need to quantify these data.

Quantification added as an extra panel.

(10) Line 164. You include H-NS in the group of "positive effectors that control fim operon expression" and you reference Ecocyc, rather than any primary reference. Nowhere in the manuscript do you mention phase variation. In the speB mutant, you see decreased fimB, increased fimE, and decreased hns expression. My interpretation of the literature suggests that this would drive the fim switch to the off-state. This could certainly explain some of the results. It is also easily measurable with PCR. This might require testing cells scraped directly from the plates.

The experiments were performed. There is no need to scrap cells from plates because the fimB result from RNAseq was from a liquid culture, and the prediction would be that the phase-locking should be evident in these cells.

(11) Figure 10. Likewise, do you know that your hns mutant is not locked in the off-state? Granted, the original hns mutants (pilG) showed increased rates of switching, but growth conditions might matter.

We also did phase variation for the hns mutant and the hns mutant was not phase locked. This result is shown. In addition to growth conditions, the strain probably matters.

(12) Line 342. You describe the total genome sequencing of W3110, yet this is not mentioned anywhere else in the manuscript.

It is now

Minor points:(13) Line 192. "One of the most differentially expressed genes...".(14) Line 202. "...implicates extracellular putrescine in putrescine homeostasis."(15) Line 209. "...potential pili regulators...".(16) You are using a variety of fonts on the figures. Pick one.(17) Figure 9A. It took me a few minutes to figure out the labeling for this figure and I was more confused after reading the legend. It would be simpler to independently label red triangles, blue triangles, red circles, and blue circles.(18) Figure 9B and 10. The reader can likely figure out what W3110_1.0_3 means, but more straightforward labeling would be better, or you need to define these labels.

All points were addressed and fixed.

**Reviewer #3 (Recommendations for the authors):**
Other comments:(1) Please go through the figures and the reference to figures in the text, as they often do not refer to the right panel (ex: figures 2 and 7 for instance). In the text, please homogenize the reference to figures (Figure 2C vs Figure 3). To help compare motility experiments between figures, please use the same scale in all figures.

This has been fixed.

(2) Lines 65-70: I am not sure I get the reason behind choosing the W3110 strain from your lab stock. In what background were the initial mutants constructed (from l.64-65)? Were the nine strains tested, all variations of W3110? If so, is the phenotype described in the manuscript robust in all strains?

We have provided more explanation. W3110 was the most stable: insertions that allowed flagella synthesis in the presence of glucose were frequent. We deleted the major flagella subunit for most experiments. Before introduction of the fliC deletion, we needed to perform experiments 10 times so that fast-moving variants, which had mutationally altered flagella synthesis, did not complicate results.

(3) Line 82-84: As stated in the public review, I think more controls are needed before making this conclusion, especially as type I fimbriae are usually involved in sessile phenotypes.

Response provided in the public response.

(4) In Figure 3: Changing the order of the image to follow the text would make the figure easier to follow.

Fixed as requested

(5) Lines 100-101: simultaneous - the results presented here do not support this conclusion. In Figure 4b, the addition of putrescine to speB mutants is actually not different from WT. From the results, it seems like one of biosynthesis or transport is needed, but it's not clear if both are needed simultaneously. For this, a mutant with no biosynthesis and no transport is needed and/or completely non-motile mutants would be needed to compare.

We disagree. If there are two pathways of putrescine synthesis and both are needed, then our conclusion follows.

(6) Lines 104-105: '... because *E. coli* secretes putrescine.' - not sure why this statement is there, as most transporters tested after are importers of putrescine? It is also not clear to me if putrescine is supplemented in the media in these experiments. If not, is there putrescine in the GT media?

Good points, and this section has been reworded to clarify these issues. Some of the material was moved to the discussion.

(7) Line 109: 'We note that potE and plaP are more highly expressed than potE and puuP...' - first potE should be potF?

This has been corrected.

(8) Figure 8: What is the difference between the TEM images in Figure 1 and here? The WT in Figure 1 does show pili without the supplementation unless I'm missing something here. Please specify.

The reviewer means Figure 2 and not Figure 1. Figure 2 shows a wild-type strain which has both putrescine anabolic pathways while Figure 8 is the ΔspeB strain which lacks one pathway.

(9) Line160-162: Transcripts for the putrescine-responsive puuAP and puuDRCBE operons, which specify genes of the major putrescine catabolic pathway, were reduced from 1.6- to 14- fold (FDR {less than or equal to} 0.02) in the speB mutant (Supplemental Table 1), which implies lower intracellular putrescine. I might not get exactly the point here. If the catabolic pathways are repressed in the speB mutant, then there will be less degradation which means more putrescine!?

Expression of these genes is a function of intracellular putrescine: higher expression means more putrescine. Any discussion of steady putrescine must include the anabolic pathways: the catabolic pathways do not determine the intracellular putrescine, they are a reflection of intracellular putrescine.

(10) Lines 162-163: Deletion of speB reduced transcripts for genes of the fimA operon and fimE, but not of fimB. It seems that the results suggest the opposite a reduction of fimB but not fimE!?

The reviewer is correct, and it is our mistake, and the text now states what is in the figure..